# Postmortem imaging reveals patterns of medial temporal lobe vulnerability to tau pathology in Alzheimer's disease

Sadhana Ravikumar [1] ✉, Amanda E. Denning[1], Sydney Lim [1], Eunice Chung [1], Niyousha Sadeghpour [1], Ranjit Ittyerah[1], Laura E. M. Wisse[2], Sandhitsu R. Das[3], Long Xie[1], John L. Robinson[3], Theresa Schuck[4], Edward B. Lee [4], John A. Detre[3], M. Dylan Tisdall[1], Karthik Prabhakaran[3], Gabor Mizsei[1], Maria Mercedes Iñiguez de Onzono Martin[5], Maria del Mar Arroyo Jiménez [5], Monica Mūnoz[5], Maria del Pilar Marcos Rabal [5], Sandra Cebada Sánchez [5], José Carlos Delgado González[5], Carlos de la Rosa Prieto[5], David J. Irwin[3], David A. Wolk[3], Ricardo Insausti[5] & Paul A. Yushkevich [1] ✉

Our current understanding of the spread and neurodegenerative effects of tau neurofibrillary tangles (NFTs) within the medial temporal lobe (MTL) during the early stages of Alzheimer's Disease (AD) is limited by the presence of confounding non-AD pathologies and the two-dimensional (2-D) nature of conventional histology studies. Here, we combine ex vivo MRI and serial histological imaging from 25 human MTL specimens to present a detailed, 3-D characterization of quantitative NFT burden measures in the space of a high-resolution, ex vivo atlas with cytoarchitecturally-defined subregion labels, that can be used to inform future in vivo neuroimaging studies. Average maps show a clear anterior to poster gradient in NFT distribution and a precise, spatial pattern with highest levels of NFTs found not just within the transentorhinal region but also the cornu ammonis (CA1) subfield. Additionally, we identify granular MTL regions where measures of neurodegeneration are likely to be linked to NFTs specifically, and thus potentially more sensitive as early AD biomarkers.

Alzheimer's disease (AD) is a slowly developing neurodegenerative disorder characterized by the accumulation of beta-amyloid plaques (Aβ) and tau neurofibrillary tangles (NFTs) in the brain, often decades before a patient becomes symptomatic. Compared to Aβ, the accumulation of NFTs is strongly correlated with neural damage and eventual cognitive decline[1–4]. According to early histological studies, NFT formation during AD follows a relatively stereotyped regional pattern of spread, with earliest cortical accumulations occurring in the medial temporal lobe (MTL)[5–13]. More specifically, the Braak staging system suggests that NFTs initially manifest in the transentorhinal cortex, which corresponds to the medial portion of Brodmann Area 35 (BA35), before spreading further into the entorhinal cortex (ERC) and eventually the cornu ammonis 1 (CA1) and subiculum subfields of the hippocampus. High levels of NFT density have also been described in regions of the amygdala closely connected to the hippocampus and ERC[8], and the stratum radiatum lacunosum moleculare (SRLM) layer of

[1]Department of Radiology, University of Pennsylvania, Philadelphia, PA, USA. [2]Institute for Clinical Sciences Lund, Lund University, Lund, Sweden. [3]Department of Neurology, University of Pennsylvania, Philadelphia, PA, USA. [4]Department of Pathology and Laboratory Medicine, University of Pennsylvania, Philadelphia, PA, USA. [5]Human Neuroanatomy Laboratory, University of Castilla La Mancha, Albacete, Spain. ✉e-mail: sadhana.ravikumar@gmail.com; pauly2@pennmedicine.upenn.edu

the CA1 subfield[12,14]. Consistent with these regions of early NFT pathology, brain atrophy measured by structural magnetic resonance imaging (MRI) in the hippocampus[15,16] and ERC[17,18] are closely correlated with cognitive decline.

Recently, with advances in granularity and accuracy of MTL subregion segmentation on in vivo MRI[19,20], a growing number of studies have focused on the more detailed interrogation of MTL subregions as a promising biomarker of AD-related neurodegeneration during the early, preclinical stages of the disease[17,19,21–24]. Early histological studies have been cited extensively in such studies and play a major role in informing the development of neuroimaging-based MTL biomarkers. However, current descriptions of the topographic distribution of NFT pathology are not sufficient for validation since they are largely based on two-dimensional (2-D) histological examination of tissue sampled at a limited number of brain locations. Therefore, our understanding of the spread of NFT pathology and the interpretation of in vivo PET and MRI biomarkers would benefit from a more complete, 3-D histological characterization of the distribution of NFT burden within the MTL.

In addition, while studies have shown that longitudinal MRI is able to detect subtle structural changes in MTL subregions associated with the earliest stages of NFT pathology accumulation[25,26], a limitation of current brain atrophy measures is that they are not necessarily specific to AD pathology. This is because multiple co-occurring pathological processes often underly neurodegeneration in patients with AD. A recent autopsy study suggested that compared to other neurodegenerative diseases, the prevalence of co-pathology is increased in AD with approximately 41–55% of individuals with AD having $\alpha$-synuclein pathology, typically associated with Lewy body dementia and Parkinson's Disease, and 33–40% of individuals having TAR DNA-binding protein 43 (TDP-43) pathology, associated with frontotemporal lobar degeneration with TDP-43 inclusions (FTLD-TDP) and Limbic-predominant Age-related TDP-43 Encephalopathy (LATE)[27]. While recent advancements in PET imaging have enabled 3-D in vivo visualization of A$\beta$ and tau pathology, there are currently no established imaging biomarkers available to specifically detect non-AD co-pathologies in vivo, although cerebrospinal fluid measures are emerging for detecting $\alpha$-synuclein pathology[28]. Previous histology and antemortem-based studies have reported that these confounding non-AD pathologies tend to follow different trajectories of MTL involvement when compared to NFT pathology[2,29,30]. This suggests that a more comprehensive postmortem characterization of NFT pathology and its specific effects within the MTL could help inform imaging biomarkers by defining regions of the MTL where neurodegeneration is most directly linked to underlying NFT pathology, as opposed to co-morbid, non-AD pathologies and aging.

In this study, we leverage foundational frameworks presented in our earlier work to construct a computational atlas of the MTL using ex vivo MRI, which enables statistical mapping of associations between MTL cortical/hippocampal thickness and multiple markers of pathology[31], and generate 3-D quantitative "heat maps" of NFT pathology from dense serial histology using machine learning algorithms[32]. Yushkevich et al.[32] showed that NFT burden measures derived from these heat maps (i.e., the mean intensity across the heat map) correlate strongly with manual tangle counts and semi-quantitative ratings of NFT severity provided by an expert neuropathologist, indicating that they accurately capture both the number and severity of tangle-like pathological inclusions in the tissue. In Ravikumar et al.[31] a computational atlas was constructed by applying a custom groupwise registration pipeline to ex vivo MRI scans of 29 human MTL specimens, and group analyses were performed correlating regional MTL thickness and semi-quantitative neuropathological ratings of tau and TDP-43 pathology. While the analyses revealed strong associations between cortical thinning and tau pathology in the

ERC and SRLM, the major limitations of that study were the pathologic heterogeneity and the relatively small size of the brain donor cohort. Here, we address these limitations by leveraging a much larger brain donor cohort to construct the atlas ($n = 55$), and only including patients with diagnoses spanning the AD continuum in the thickness analyses ($n = 47$). Furthermore, we improve upon the atlas by incorporating cytoarchitecture-guided segmentations of MTL subregions from a larger number of cases ($n = 17$) and including the full extent of the SRLM layer. In this study we additionally expand on the group-level analyses presented in Yushkevich et al.[32], which used 3-D quantitative maps of NFT burden derived from serial histology in 15 brain donors with heterogeneous neuropathological diagnoses, to describe the average distribution of NFT burden in the space of an in vivo brain template. Here, we examine 3-D NFT burden maps from 25 cases, with diagnoses on the AD continuum (inclusion/exclusion criteria described in the results section), now in the space of our high-resolution, ex vivo MRI atlas; and we compare differences in the 3-D topography of NFT pathology between early and late Braak stages. Unlike in[32], where the anatomical labels for the analysis were derived from in vivo atlases, here we use histologically-defined subregion boundaries to describe anatomical differences in NFT burden. In addition to enabling a more granular and accurate 3-D characterization of the distribution of NFT pathology across specimens, using the ex vivo MRI atlas permits, for the first time, regional thickness analyses directly linking quantitative NFT burden measures and cortical thickness. Taken together, these methodological improvements and the better selection of brain donors create a far more precise postmortem reference which can be used to validate and inform the development of future in vivo imaging biomarkers and studies investigating the spread of tau pathology in early AD.

## Results

### Brain donor cohorts

Brain hemisphere specimens from 55 donors, aged 57-99 years were obtained from autopsy cases at the Human Neuroanatomy Laboratory at the University of Castilla-La Mancha (HNL, $n = 21$) and the Center for Neurodegenerative Disease Research at the University of Pennsylvania (CNDR, $n = 34$). Donors from CNDR were participants in vivo ageing and dementia research and included patients from the Penn Frontotemporal Degeneration Center and the Penn Alzheimer's Disease Research Center. For this study, we were interested in characterizing neurodegeneration in patients with a primary diagnosis of Alzheimer's disease pathology. Therefore, while all 55 cases were used to construct a computational atlas of the MTL, clinical analyses were performed in a subset of 47 cases with an "AD continuum" diagnosis and no confounding non-AD tau (i.e., Pick's disease, progressive supranuclear palsy, and corticobasal degeneration) or FTLD-TDP43 pathology. AD continuum diagnoses include a neuropathological diagnosis of 'unremarkable brain', primary age-related tauopathy and AD neuropathologic change, and co-pathologies such as Lewy body dementia, cerebrovascular disease, multiple system atrophy, limbic age-related TDP-43 encephalopathy, hippocampal sclerosis, and cerebral amyloid angiopathy. 3-D maps of quantitative NFT burden were constructed for a smaller subset of 25 "AD continuum" cases with serial anti-tau immunohistochemistry (IHC) data. Table 2 provides a summary of demographic and diagnostic data for the AD brain donor cohorts, with additional details of the full dataset in Supplementary Table 1. Note that in brains with co-occurring pathologies, the neuropathology that is more dominant or advanced is listed as the primary neuropathological diagnosis. However, there are cases where it is not clear which neuropathology is more dominant. In such cases, the driving neuropathology that is more responsible for the clinical phenotype is listed as the primary neuropathological diagnosis even though both the primary and secondary neuropathologies are equally severe.

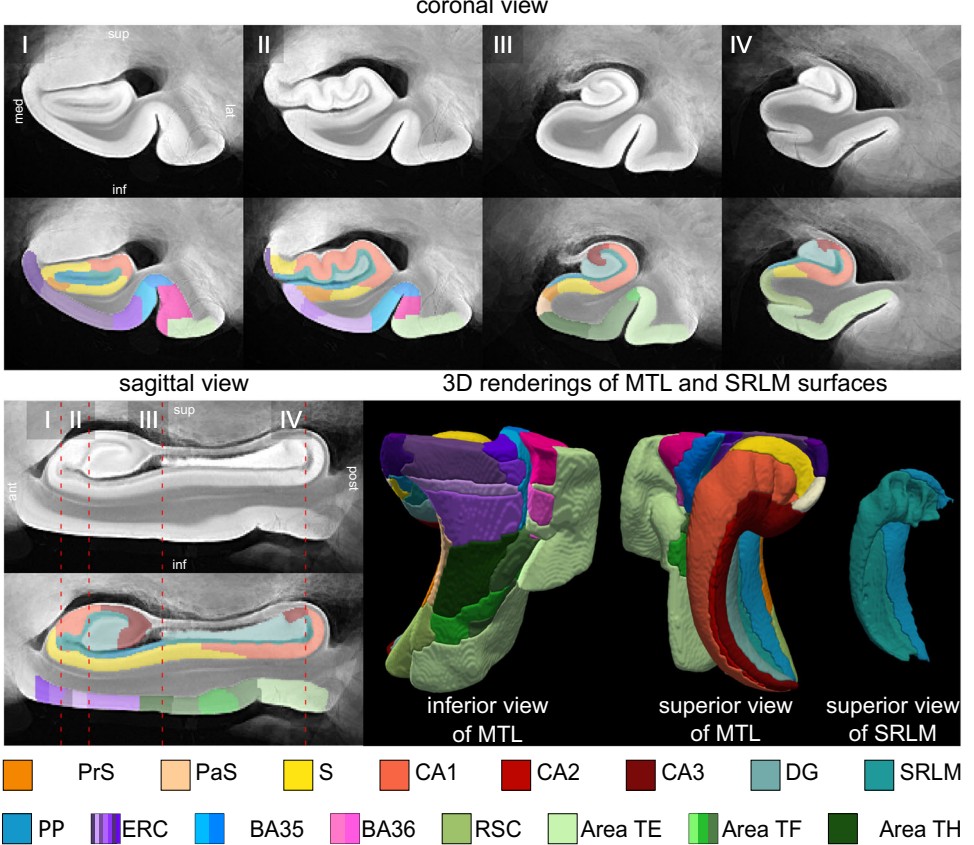

**Fig. 1 | Computational atlas of the medial temporal lobe (MTL) constructed from ex vivo MRI scans of 55 donor specimens and serial histology in 17 specimens.** Four coronal sections are shown ordered from anterior (ant) to posterior (post), indicated as I, II, III, and IV, as well as a sagittal cross-section and 3-D reconstructions of the MTL and SRLM surfaces. For each cross-sectional view, the "average" MRI is shown with and without the histology-derived, consensus MTL subregion segmentation. The subregion labeling includes the subdivisions of the ERC (in shades of purple), BA35 (in shades of blue), BA36 (in shades of pink), and area TF (in shades of green). (med medial, lat lateral, sup superior, inf inferior, PrSPaS pre/parasubiculum, S subiculum, CA cornu ammonis, DG dentate gyrus, SRLM stratum radiatum lacunosum molecular, PP perforant pathway, ERC entorhinal cortex, BA Brodmann Area, RSC retrosplenial cortex).

## 3-D distribution of NFT burden within the MTL

Figure 1 shows the MRI atlas of the MTL constructed from 55 ex vivo specimens as a synthetic "average" MR image and a consensus MTL subregion segmentation derived from serial histology in 17 specimens. The consensus segmentation provides a detailed visualization of 26 MTL subregion labels, including the 7 subdivisions in the ERC, 2 subdivisions each in BA35 and BA36, and 2 subdivisions in Area TF (Table 1). The atlas construction pipeline combines specimens obtained from both left and right hemispheres (by flipping the latter), resulting in a single atlas that describes MTL anatomy and pathology bilaterally.

For a subset of 25 "AD continuum" cases, we reconstructed 3-D quantitative NFT burden maps from serial anti-tau IHC sections and brought them into the space of the MTL atlas for group-level analyses. Figure 2A plots the average NFT burden maps computed separately for specimens with a low B score (B0 or B1, which corresponds to Braak stages 0–II, 11 cases) and a high B score (B2 or B3, which corresponds to Braak stages III–VI, 14 cases). Figure 2A also includes four frequency maps for the low and high Braak groups, where each of the four frequency maps indicates how frequently among the specimens included in the subgroup the NFT burden exceeds a certain threshold. Regions of the MTL with a high frequency are indicative of anatomical regions affected during the early stages of the disease. The coronal cross-sectional views of the maps reveal a high average NFT burden in regions corresponding to the CA1 subfield of the hippocampus, and the ERC, and BA35 regions of the parahipppocampal cortex during the

early Braak stages. Interestingly, in the ERC and BA35, a higher NFT burden appears to be concentrated in the outer cortical layers. We are now able to better observe this detailed pattern because of our higher-resolution mapping. At later Braak stages, the average NFT burden is visibly higher, with regions of high burden spreading further into the subiculum, ERC, BA36, and more posteriorly into Areas TF-TH. In 2-D histological studies, sectioning is typically done in the coronal plane, thus limiting our knowledge of the distribution of NFTs along the anterior-posterior axis of the MTL. The sagittal views of our 3-D mappings reveal a marked anterior to posterior gradient in NFT burden along the parahippocampal gyrus, visible in cases at both the early and late Braak stages. We also observe an increased NFT burden in CA1 along the full length of the hippocampus. These patterns are clearly depicted in 3-D visualizations comparing the average NFT burden maps at the early and late Braak stages, as shown in Supplementary Movie 1.

These differences in the distribution of NFT burden across anatomical subregions and Braak stages are quantified in Fig. 2B, C. Quantitative summary measures of NFT burden were computed for each anatomical region of interest (ROI) defined in the histology-derived MTL subregion segmentation. We combined the presubiculum and parasubiculum into a single ROI and did the same for the subdivisions within the ERC, those within BA35, and those within BA36. Figure 2B shows box and whisker plots for each ROI comparing the distribution of mean NFT burden between cases in the low and high Braak groups, and Fig. 2C plots the distribution of mean NFT burden in

**Table 1 | Abbreviations and descriptions for the anatomical subregions defined within the medial temporal lobe atlas**

| Abbreviation | Description |
|---|---|
| PrS | Presubiculum |
| PaS | Parasubiculum |
| S | Subiculum |
| CA1-3 | Cornu Ammonis 1-3 |
| DG | Dentate gyrus |
| SRLM | Stratum radiatum lacunosum moleculare. |
| PP | Perforant pathway. Refers to the white matter of the subiculum. |
| ERC | Entorhinal cortex. This region contains several subdivisions detailed in Fig. 6 |
| BA35 | Brodmann Area 35 or perirhinal cortex. This region contains oblique and dorsal subdivisions. |
| BA36 | Brodmann Area 36 or perirhinal cortex. This region contains rostral and caudal subdivisions. |
| RSC | Retrosplenial cortex |
| Area TE | Also known as the inferotemporal cortex, this subregion is laterally adjacent to the parahippocampal cortex. Although subdivisions of area TE have been identified, here we group them all under a single label. |
| Area TF | Subdivision of the parahippocampal cortex based on cytoarchitectural features. This region contains medial and lateral subdivisions. |
| Area TH | Subdivision of the parahippocampal cortex based on cytoarchitectural features. |

each ROI computed across all 25 cases normalized to the mean NFT burden for BA35. We chose BA35 as the reference region for normalization as it corresponds to the transentorhinal region, described as the earliest cortical region affected by tau pathology in AD[11]. Significant or near significant (subiculum: $p = 0.051$) increases in the distribution of NFT burden between the low and high Braak cases are observed across all anatomical ROIs. Consistent with the qualitative findings, the highest levels of NFT burden relative to BA35 are found in CA1 and the ERC, and the lowest levels are observed in the pre- and parasubiculum region, Area TE, and dentate gyrus.

**Association of quantitative tau pathology measures with MTL cortical thickness**

In the subset of cases with serial anti-tau IHC, we performed thickness analyses using summary measures of NFT burden derived ipsilaterally from the quantitative NFT burden heatmaps. First, to investigate the local effects of tau burden on cortical thinning, we examined the Spearman rank correlation between mean NFT burden and mean thickness computed within the same anatomical ROI. The two exceptions were SRLM and perforant pathway, since these regions do not contain neurons and thus are not expected to accumulate NFT pathology. Instead, SRLM thickness was correlated with the mean NFT burden in the neighboring CA1 subfield and perforant pathway thickness was correlated with the mean NFT burden in the structurally connected ERC region. Scatter plots for the different MTL subregions are shown in Fig. 3. Negative correlations are observed across all subregions, although only the perforant pathway, dentate gyrus, ERC, pre/parasubiculum and the posterior MTL subregions (i.e., Areas TE, TF, and TH) reach significance. Trend level associations are observed in the subiculum and BA36 (uncorrected $p < 0.1$). Only NFT-structure associations in dentate gyrus, Area TE and Area TH remained significant when age was added to the model and no significant associations were observed when both age and sex were added to the model, likely due to the small sample size. The scatter plots show high levels of variability in cortical thickness measurements across specimens even at low levels of NFT burden, suggesting that non-AD developmental or aging-related variations in thickness across cases may be weakening

the detected associations. In Supplementary Fig. 3, we include scatter plots showing the relationship between age and mean NFT burden computed within each of the anatomical ROIs. Significant correlations between age and NFT burden are observed across all subregions ($R > 0.53$ across all ROIs). Since age is likely in the causal pathway between the accumulation of NFT burden and neurodegeneration, with increased age leading to higher levels of tau pathology, which in turn leads to reduced cortical thickness, by correcting for age, we may be obscuring a crucial aspect of the pathway that we are interested in investigating, rather than correcting for a confounder. In these analyses, we did not co-vary for TDP-43 or α-synuclein pathologies. However, all the cases in this subset have an average TDP-43 pathology rating that is less than 1, with the majority of the cases having no TDP-43 pathology (see Table 2).

We also performed pointwise thickness analyses to visualize the regional atrophy patterns obtained in correlation with quantitative NFT burden measures derived from different anatomical subregions (Fig. 4). More specifically, we correlated regional thickness with the mean NFT burden computed within ROIs involved in the early stages of the disease (i.e., CA1, BA35, and the ERC), and late stages of the disease (i.e., pre/parasubiculum and Area TE). These ROIs were chosen based on the analyses shown in Fig. 2C which examined the relative distribution of NFT burden across the different MTL subregions. When using NFT burden measures derived from ROIs affected during the later stages of the disease, we observe significant correlations between NFT burden and cortical thickness in the ERC, extending into BA35, the CA1/subiculum region, and SRLM. The strength of pathology/thickness associations in these regions progressively weakens as we use NFT burden measures derived from regions affected earlier in the disease process (i.e., CA1, BA35, and ERC), with only SRLM atrophy patterns observed in association with CA1 NFT burden. The weakened associations between cortical thickness and mean NFT burden measured in these early regions are likely because these regions have reached a ceiling of high NFT burden across cases. Supplementary Fig. 4 shows the results of the same analysis, repeated with age and sex included as covariates in the model. We observe weakened results, with only atrophy patterns in the CA1/subiculum region remaining statistically significant when using NFT burden measures derived from late tau subregions.

In a supplementary analysis, we repeated the pointwise thickness analysis in this subset of 25 cases using the semi-quantitative tau ratings derived from the MTL contralateral to the thickness measures (see Supplementary Fig. 5). Overall, we find weakened tau-thickness associations with small clusters of correlation observed only in the SRLM ($p = 0.009$) and CA1 ($p = 0.051$) subfield of the hippocampus. While the MTL t-statistic map shows stronger pointwise correlations in the ERC region, they do not reach significance at a cluster level. Bearing in mind differences in protocols and staining used to obtain the ipsilateral and contralateral measurements, the stronger atrophy patterns observed when using the ipsilateral measurements highlight the value of using 3-D, quantitative ratings of NFT pathology specifically.

**Association of semi-quantitative measures of different neurodegenerative pathologies with MTL cortical thickness**

In the larger dataset of 47 specimens, we studied the association between pointwise regional thickness of the MTL cortex, hippocampal gray matter, and SRLM, and semi-quantitative, histologic ratings of tau, TDP-43, and α-synuclein pathologies derived from the MTL contralateral to the thickness measures. The resulting statistical maps are shown in Fig. 5. The correlation analysis between tau pathology and thickness, with only age and sex as co-variates, reveals clusters of significant associations (after correction for multiple hypothesis testing[33]) between increased tau severity and cortical thinning in the ERC region and SRLM. Our analysis including TDP-43 and α-synuclein pathologies in the model as nuisance covariates resulted in similar

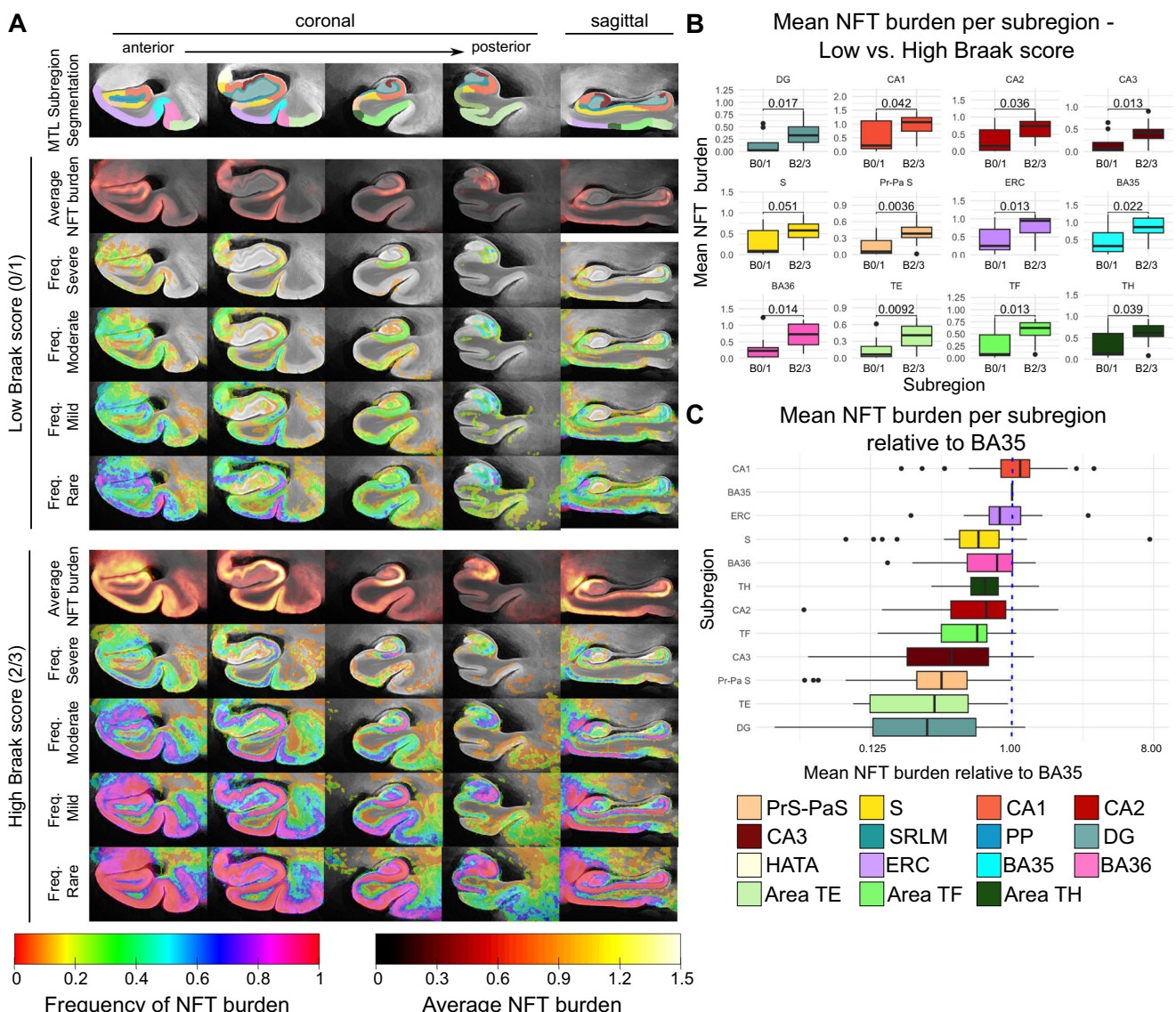

**Fig. 2 | Characterization of histology-derived quantitative tau neurofibrillary tangle (NFT) burden maps in the space of a 3-D ex vivo anatomical atlas.**
**A** Average and summary frequency maps of NFT burden in the space of the ex vivo MRI atlas of the medial temporal lobe (MTL). Maps are computed separately for specimens with a low B score (B0 or B1, which corresponds to Braak stages 0–II; *n* = 11) and a high B score (B2 or B3, which corresponds to Braak stages III–VI). For each subgroup, four coronal and one sagittal cross-sectional view of the average NFT burden map, and four frequency maps are visualized. The frequency maps at each voxel describe the fraction of cases for which the NFT burden at that voxel was above a given threshold. Thresholds were chosen based on the analysis conducted by Yushkevich et al. (2021) and correspond to different levels of pathological burden (>1.0 for `severe`; >0.5 for `moderate`; >0.25 for `mild`; >0.1 for `rare`). The top row shows the corresponding cross-sections of the consensus histology-based MTL subregion segmentation. For simplicity, we combine the presubiculum and parasubiculum labels, and the subdivisions of the ERC, BA35, and BA36. **B** Box

plots comparing the distribution of mean NFT burden within each subregion between patients with a low and high B score. Using the two-sided *t* test, significant increases in NFT burden are observed in all subregions except the subiculum, where it nearly reaches significance (*p* < 0.05). **C** Box plot showing the NFT burden in MTL subregions normalized to BA35 NFT burden (dashed blue line). Subregions are sorted in order of decreasing mean NFT burden relative to BA35, going from top to bottom. Box plots in (**B**) and (**C**) show the median as the middle box line, first quartile (Q1) and third quartiles (Q3) as box edges (denoting the interquartile range, IQR), whiskers as the minima/maxima and outliers based on thresholds < Q1 − 1.5(IQR) or > Q3 + 1.5(IQR). Sample sizes are provided in Supplementary Table 2. Source data for 2B) and (C) are provided as a Source Data file. (S subiculum, PrS-PaS Pre/Parasubiculum, SRLM stratum radiatum lacunosum molecular PP perforant pathway, CA cornu ammonis, DG dentate gyrus, HATA hippocampal amygdala transition area, ERC entorhinal cortex, BA Brodmann area).

atrophy patterns, with stronger associations observed in the ERC, and an additional significant cluster seen along the border of the subiculum and CA1 regions. To further understand the neurodegenerative effects of TDP-43 and α-synuclein, we repeated this analysis with TDP-43 and α-synuclein as the variables of interest instead and found significant associations between TDP-43 pathology and thickness in the subiculum and CA1 region, spanning the full length of the hippocampus. No significant associations were observed between α-synuclein pathology and thickness, consistent with previous findings[2,34].

### 3-D Distribution of NFT burden within the ERC
In an exploratory analysis, we were interested in analyzing variability in the distribution of NFT burden within the subfields of the ERC. In[35], Insausti et al. define eight distinct subfields within the human ERC based on cytoarchitectural features: Eo (olfactory), ER (rostral), EMI (medial intermediate), EI (intermediate), ELr (lateral rostral), ELc (lateral caudal), EC (caudal), and ECL (caudal limiting). Figure 6A shows the histology-based consensus labeling of these entorhinal subfields in the space of the MTL atlas at different cross-sectional levels from

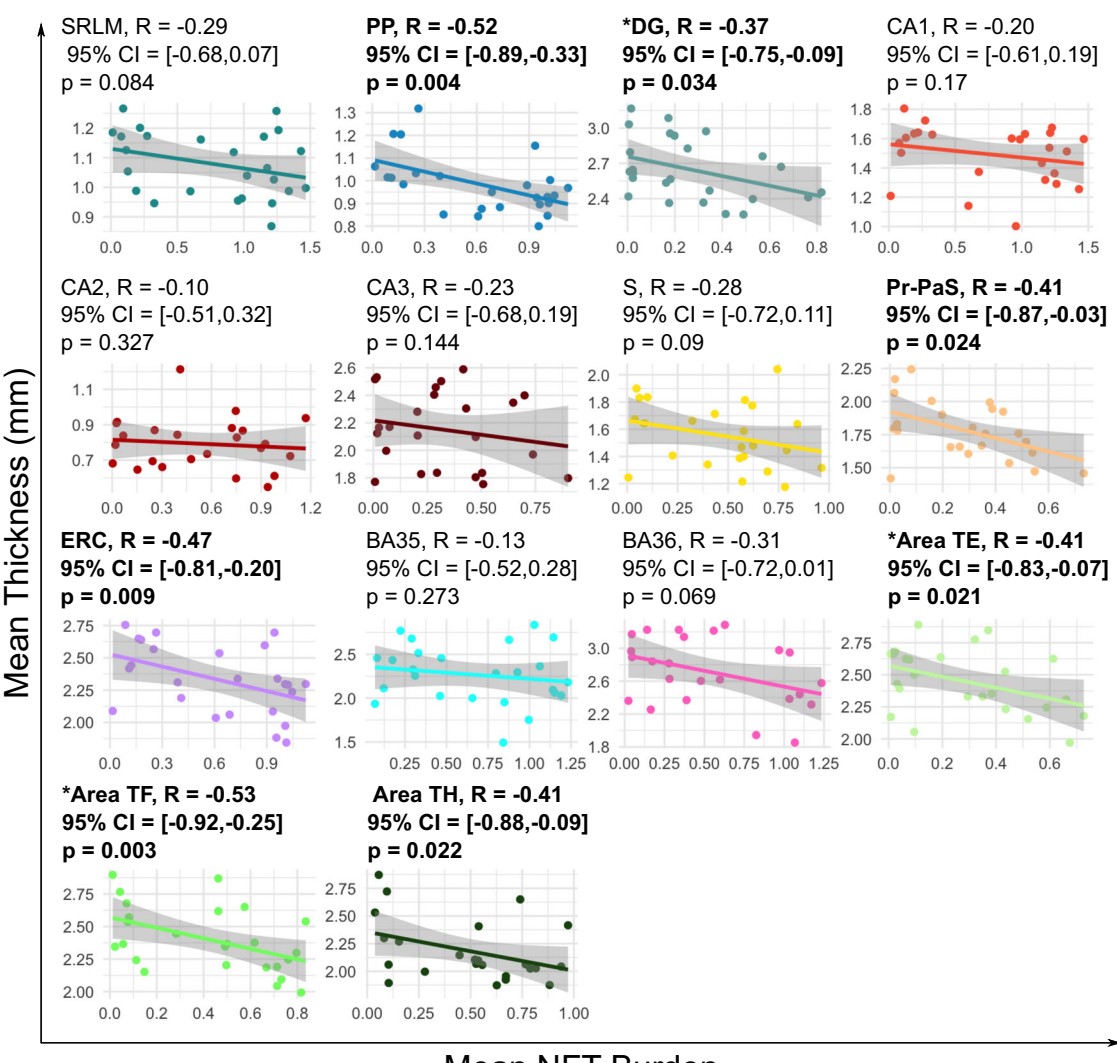

**Fig. 3 | Relationship between mean cortical thickness and mean quantitative NFT burden computed within the same MTL subregion.** The scatter plots illustrate the regional relationship between cortical thickness and NFT burden measured within the same subregion for each of the 14 MTL subregions. Each plot also includes the Spearman's rank correlation calculated between mean cortical thickness and mean NFT burden within the same subregion. The two exceptions are SRLM and PP since these ROIs do not directly accumulate NFT pathology. SRLM and PP thickness are correlated with mean NFT burden in CA1 and the ERC region respectively. Significant negative associations are bolded (one-sided, uncorrected *p* < 0.05). The asterisk is used to indicate ROIs where the model including both age and mean NFT burden is significant. Sample sizes are provided in Supplementary Table 2. Source data are provided as a Source Data file. (*S* subiculum, PrS-PaS: Pre/Parasubiculum, SRLM stratum radiatum lacunosum molecular, PP perforant pathway, CA cornu ammonis, DG dentate gyrus, ERC entorhinal cortex, BA Brodmann area).

anterior to posterior. Note that since the anterior border of the atlas only starts at the hippocampal head, we are partially missing the anterior extents of the entorhinal subfields Eo, ER, and EMI. In our analysis, we exclude quantitative measurements derived from Eo and ER since much of the extent of these subfields is not included in the atlas. Figure 6B plots the distribution of mean NFT burden in each of the entorhinal subfields, computed across all cases and separately for cases with low and high B scores respectively. These plots are shown together with corresponding surface heat maps which highlight the distribution of NFT burden across the entorhinal subfields ranked in order of highest to lowest mean NFT burden. In the low Braak group, the highest NFT burden is observed in EMI, followed by ELr and ELc. In high Braak cases, the highest NFT burden is observed in ELc, followed by EI and ELr. Overall, our results indicate that the anterior-lateral subfields of the ERC may be more vulnerable to NFT pathology, with the lowest NFT burden observed in EC and ECL. Lastly, we performed Spearman rank correlations between mean cortical thickness and

mean NFT burden computed within each entorhinal subfield (Fig. 6C), and found negative correlations across all subfields, although only the associations in EC, ECL, and ELr reach significance.

## Discussion

Leveraging postmortem imaging of a relatively large number of human MTL specimens with neuropathological diagnoses on the AD continuum, we characterize at an unprecedented level of detail, the 3-D probabilistic distribution of NFT burden at the different stages of AD and the regional effects of tau pathology on MTL neurodegeneration. This allows us to visualize and analyze patterns of NFT distribution along both the coronal and sagittal axes of the MTL and thus offers more extensive information than current histology-based descriptions of NFT topography in AD, which are in 2-D and based on selective sampling of the MTL[6,8,9,11]. According to the Braak staging system, NFT pathology initially accumulates in the border region between the 'transentorhinal cortex', the medial portion of BA35, and the lateral

**Table 2 | Demographic and diagnostic summaries of the brain donor cohort with Alzheimer's disease**

|  | Full AD Cohort | AD Cohort subset with NFT Burden Maps |
|---|---|---|
| N | 47 | 25 |
| Age | 78.6 ± 10.1 (57–99) | 78.6 ± 10.9 (59–97) |
| Sex | 29M/18F | 16M/9F |

| Contralateral MTL Pathology Rating |  |  |
|---|---|---|
| Average Tau rating | 1.55 ± 0.82 (0–3) | 1.51 ± 0.80 (0.33–3) |
| Average A rating | 1.14 ± 0.99 (0–3) | 1.15 ± 0.98 (0–3) |
| Average TDP-43 rating | 0.16 ± 0.45 (0–2.33) (81% ratings = 0) | 0.08 ± 0.20 (0–0.67) (84% ratings = 0) |
| Average a-synuclein rating | 0.27 ± 0.61 (0–2.33) (79% ratings = 0) | 0.22 ± 0.64 (0–2.33) (88% ratings = 0) |

| Neuropathological Diagnosis (from contralateral sampling) | Primary | Secondary | Primary | Secondary |
|---|---|---|---|---|
| Unremarkable Brain | 5 | 0 | 2 | 0 |
| Primary Age Related Tauopathy | 2 | 2 | 2 | 0 |
| Low ADNC | 11 | 3 | 8 | 0 |
| Intermediate ADNC | 10 | 3 | 7 | 2 |
| High ADNC | 10 | 1 | 3 | 1 |
| Lewy Body Disease | 7 | 7 | 3 | 2 |
| Cerebrovascular Disease | 0 | 9 | 0 | 5 |
| LATE | 2 | 2 | 0 | 1 |
| Multiple Systems Atrophy | 1 | 0 | 0 | 0 |
| Hippocampal Sclerosis | 0 | 1 | 0 | 0 |
| **Neuropathological Staging** | **0** | **1** | **2** | **3** | **0** | **1** | **2** | **3** |
| A (Amyloid) | 8 | 10 | 11 | 18 | 4 | 6 | 5 | 10 |
| B (Braak) | 4 | 17 | 15 | 11 | 1 | 10 | 10 | 4 |
| C (CERAD) | 16 | 6 | 9 | 16 | 8 | 5 | 4 | 8 |

This includes medial temporal lobe (MTL) pathology ratings, primary and secondary postmortem diagnoses, and global neuropathological staging using the Hyman et al. (2012) protocol. The tau, TDP-43, and α-synuclein ratings for each specimen are averaged across three MTL locations (*ERC* Entorhinal cortex, *CA* Cornu Ammonis and *DG* Dentate Gyrus) sampled from the contralateral hemisphere. The ratings range from 0 (none) to 3 (severe). (*ADNC* Alzheimer's disease neuropathologic change, *LATE* Limbic Age-related TDP-43 Encephalopathy).

portion of the ERC (ELr and ELc) during Stages I and II (corresponding to B1) followed by the CA1 and subiculum subfields of the hippocampus (Stages II/III). Here, in addition to early transentorhinal involvement, we observe similar and in some cases higher levels of NFT burden in the CA1 subfield of the hippocampus, suggesting a more widespread distribution of NFT pathology during this early stage. We also observe greater vulnerability to NFT pathology in the anterior portion of the parahippocampal gyrus extending towards the temporal pole. This increased anterior involvement includes the region that appears to correspond to the amygdala, consistent with findings reported in previous neuropathology studies that have shown amygdala changes due to the presence of NFT pathology[8,10,36,37]. Overall, these findings add important histological evidence showing the broader impact of NFT pathology during early AD beyond just the transentorhinal region, highlighting the need for examining the hippocampal subfields and amygdala in future in vivo studies of early AD.

The 3-D mapping of NFT burden presented in this work expands on the recent study conducted by Yushkevich et al. in which 3-D NFT burden maps, generated in 15 specimens, were used to characterize the topographic distribution of NFT burden in the space of a coarser in vivo brain MRI template[32]. Compared to the preliminary results presented in[32], the current work uses a larger and better-defined set of AD cases, thus enabling more expansive analyses focused on the MTL

and characterization of NFT burden in relation to disease stage. In Supplementary Fig. 6, we map our ex vivo atlas to the in vivo brain template used in[32] to provide a side-by-side comparison of the 3D mapping presented in the current work and[32]. Although the distribution of NFT pathology observed in the current dataset is mostly consistent with the pattern of distribution reported in[32], we see that by leveraging a more advanced, shape-based atlas construction pipeline and cytoarchitecture-guided MTL subregion segmentations derived from serial histology, the current mapping provides a fine-grained visualization of the differential involvement of NFT pathology across the MTL that is also more precisely linked to the specific subregion boundaries. While in[32], an anterior to posterior gradient in NFT distribution was observed in both the parahippocampal gyrus and hippocampus, here our average maps suggest greater NFT accumulation in the anterior parahippocampal gyrus but more extensive CA1 involvement extending to include both the anterior and posterior regions of the hippocampus, even during the early Braak stages. This difference could also be indicative of some Braak III cases being misclassified as Braak Stage II due to NFT pathology not being present or missed in the CA 1/subiculum region of a single histology slice sampled from the opposite hemisphere during standard autopsy.

Other studies have focused on developing frameworks for reconstructing 3-D mappings of tau pathology in the brain using ground truth information from histology[38,39], although in a limited number of specimens. Ushizima et al. present an end-to-end pipeline for creating 3-D tau mappings across the whole brain in the space of 1 mm isotropic ex vivo MRI, with the goal of validating in vivo molecular imaging[39]. While the increased scope of brain regions is advantageous, the pipeline offers a lower spatial resolution and has only been applied to two specimens. More similar to the current work, Stouffer et al. focus on reconstructing the 3-D distributions of NFT density within the MTL of two advanced AD specimens, and link pathology measures to longitudinal atrophy rates measured within the amygdala, and transentorhinal and entorhinal cortices[36]. In support of our findings[36], report significant atrophy rates in the ERC, and the highest levels of NFT burden in the ERC and amygdala. Compared to both these prior studies, the current work presents a more comprehensive analysis of 3-D NFT burden maps generated from a large number of both early and late AD cases.

Importantly, by constructing an ex vivo atlas, we are now able to perform histology-based analyses linking local NFT burden and neurodegeneration. The results of our pointwise thickness analyses, using both quantitative and semi-quantitative NFT burden measures, further highlight the early effects of NFT pathology on neurodegeneration in the ERC, and CA1 and subiculum subfields of the hippocampus. We also observe significant associations in the SRLM, in agreement with histopathology studies showing early involvement of tau pathology in the SRLM of CA1, as well as prior in vivo MRI studies which have demonstrated SRLM atrophy in patients with AD[12,40–42]. The clusters identified in the current work are in regions consistent with our initial findings[31], and as expected, stronger in absolute terms. Similar to our earlier work, we still do not observe any tau effects in BA35. Indeed, the average NFT burden maps reveal a high NFT burden in BA35, suggesting that noise in the thickness measurements obtained from this region and registration errors are likely contributing to the weakened associations. This further motivates future work focused on developing more advanced groupwise registration techniques to better handle anatomical normalization of highly variable sulcal patterns. In our analysis using semi-quantitative neuropathology ratings, when comparing atrophy patterns in relation to tau pathology versus TDP-43 pathology, we observe that TDP-43 pathology has a much greater effect on the pyramidal layers while tau is more correlated with the SRLM. This is in line with the CA1 subfield showing robust atrophy, particularly with hippocampal sclerosis associated with TDP-43 pathology[43]. In[34], Wisse et al. found no significant associations when

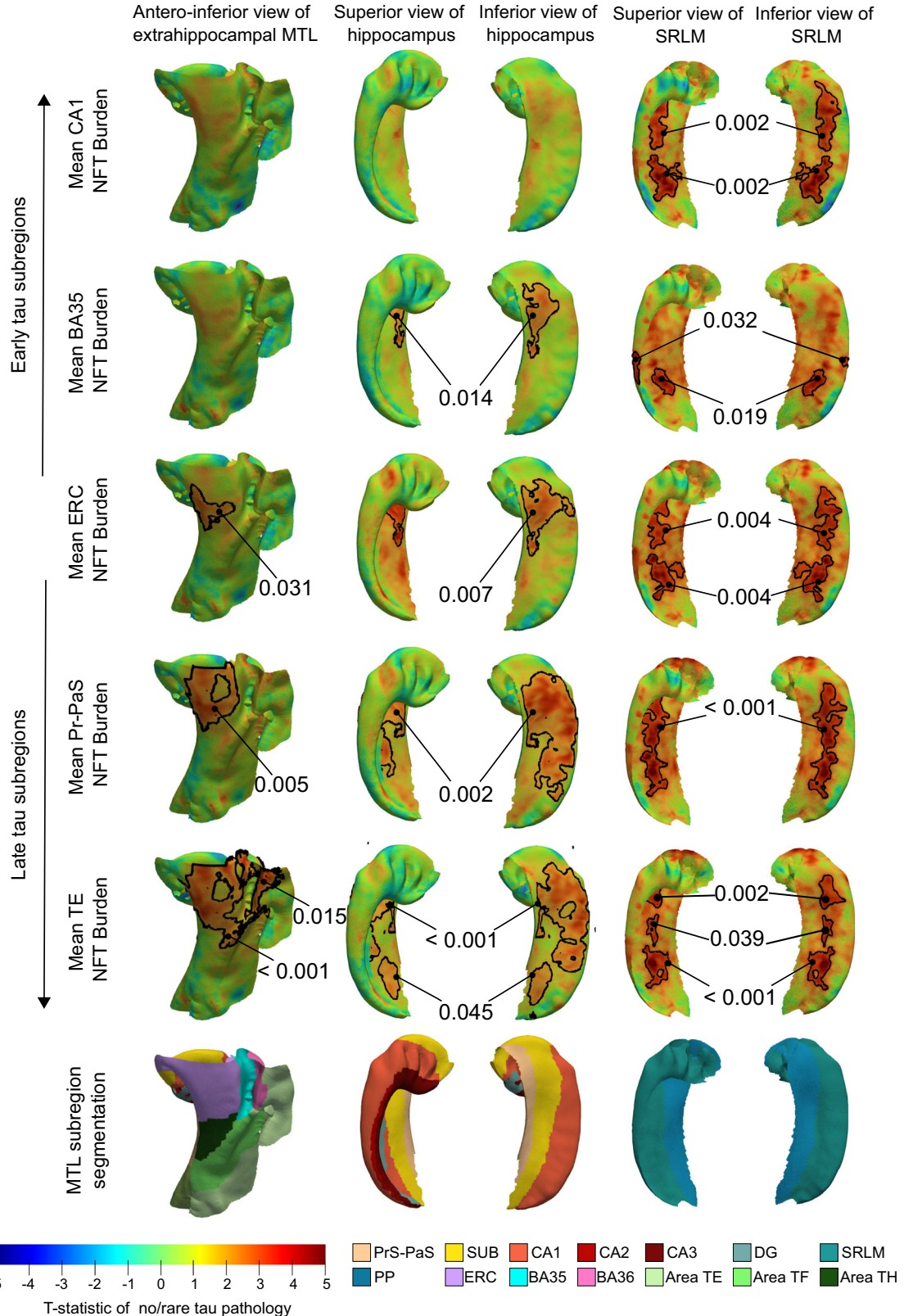

investigating synergistic effects between tau and TDP-43 on MTL neurodegeneration. This analysis was done in a slightly larger dataset, albeit with more variable neuropathological diagnoses. Future analyses of a larger AD dataset including more cases with severe TDP-43 pathology would enable further investigation of interactive effects between tau and TDP-43 pathology. Together, the results of our regional thickness analyses suggest that the ERC and SRLM might be more specific to tau pathology, as opposed to aging and non-AD pathologies. We hypothesize that by mapping the detected 'hotspots' to the in vivo domain, in vivo measures of structural change derived from these regions would provide heightened sensitivity and utility over conventional whole hippocampus imaging biomarkers.

**Fig. 4 | Association between pointwise medial temporal lobe thickness and ipsilateral, quantitative NFT burden measures.** The t-statistic maps show the correlation between pointwise cortical thickness and the mean NFT burden computed within different anatomical subregions ($n = 25$). The NFT burden measures are computed based on the 3-D quantitative NFT density maps. The analyses are arranged in the order in which the anatomical subregions used are affected during the AD process, from early to late, based on the results shown in Fig. 2C. The t-statistic maps reveal increasingly strengthened associations with NFT burden measures derived from regions affected during the later stage of AD. Clusters were defined based on an empirical threshold (uncorrected $p < 0.01$) and permutation testing with the Freedman & Lane method (1000 iterations) was used to assign each potential cluster a one-sided, corrected $p$-value. To account for multiple comparisons, the analysis uses cluster-level family-wise error rate correction. The clusters outlined in black indicate regions where a significant correlation was observed after correction for multiple hypothesis testing (corrected $p < 0.05$). No covariates were included in this model. (S subiculum, PrS-PaS Pre/Parasubiculum, SRLM = stratum radiatum lacunosum molecular, PP perforant pathway, CA cornu ammonis, DG dentate gyrus, ERC entorhinal cortex, BA Brodmann area).

At this high resolution, we are now able to further investigate the ERC and describe the accumulation of NFT pathology across its subfields. There has been increased interest in characterizing atrophy patterns and pathology burden across the subfields of the ERC due to its involvement in early, preclinical AD[36,44–46]. However, with few approaches offering the high spatial resolution required for such analyses, our understanding of ERC vulnerability at the level of substructures is still limited. Here, we assess the distribution of NFT burden across the entorhinal subfields using histopathological ground truth and find higher levels of NFT burden in EI, ELr, and ELc, which correspond to the anterior-lateral parts of the ERC. This makes sense given that the anterior-lateral ERC borders with the transentorhinal cortex, a site of early NFT pathology[5]. We note that our result of higher anterior NFT burden in the ERC of low Braak cases appears inconsistent with the recent postmortem study by Llamas-Rodríguez et al. that characterized NFT burden in 10 preclinical AD cases (age range 59–84 years), and found higher tau in the posterior-lateral subfields: ECL, ELc and ECs (lateral portion of ERC), and increased anterior tau in older individuals (age range 75–84 years)[44]. Indeed, the patient population used in the present study includes many older participants which could explain the high anterior tau burden observed even in the cases with low B scores (B0/1 subset: age range 59–93 years). While this is a relatively small sample size to reliably study age effects, this finding together with our result showing weakened pointwise tau-thickness associations in the ERC when accounting for age and sex effects (see Supplementary Fig. 4), highlights the importance of future studies examining age-related spreading of tau pathology, particularly within the ERC. The patterns of tau-thickness correlations observed in our analyses using semi-quantitative tau ratings, which include age and sex in the model, are consistent with the atrophy patterns detected in the analyses using quantitative NFT burden measures, suggesting that these neurodegenerative effects are largely driven by disease severity and not age-related factors. Furthermore, even though we observe a strong association between age and NFT burden in our dataset, we also observe significant increases in NFT burden across all MTL subregions when comparing cases at low and high Braak stages. In the present study, we don't have a large enough sample size to investigate age-related pathology patterns and analyze the primary AD cases separately from other groups such as PART and those where AD is the secondary diagnosis. Our findings motivate future analyses using a larger dataset to perform tau mediation analyses and examine potential differences in 3D tau distributions and pathology-structure associations between diagnostic groups and age groups.

Our finding of higher NFT burden in the anterior-lateral ERC also makes sense in the context of structural connectivity patterns within the MTL. The lateral part of the ERC receives the heaviest projections from the perirhinal cortex and is a zone of convergence of polysensory association cortex inputs[47]. In contrast, more medial regions of the ERC receive stronger projections from the parahippocampal and other sources as well[48]. While both regions, in turn, send connections to the subiculum/CA1 subfields of the hippocampus, studies have demonstrated that there is substantial segregation between the two pathways resulting in two distinct cortical networks within the MTL, namely the anteriortemporal (AT) system, which includes the perirhinal cortex

and amygdala, and the posterior-medial (PM) system, which includes the parahippocampal cortex[49,50]. The higher NFT burden and atrophy patterns we observe in the anterior parahippocampal gyrus, more specifically the anterior-lateral part of the ERC, and consequently the subiculum/CA1 region, which receives projections from the ERC, could therefore be explained by the early involvement of the perirhinal cortex (specifically the BA35 subregion) in AD. This is in line with recent studies suggesting that the AT network is more affected by tau deposition during the earliest stages of AD[46,51–53]. This result suggests that tau PET or MRI measures of the anterior-lateral ERC might be more sensitive to detect early changes in AD in contrast to the now commonly used ERC. Current studies examining the role of structural and functional connectivity in the spread of tau pathology typically use the ERC as a seed for connectivity analysis, with definitions based on in vivo segmentation protocols and tau-PET[54,55]. Recent studies have explored more granular analyses associating patterns of tau deposition with connectivity networks derived from the anterior-lateral and posterior-medial ERC[54,56]. In[56], Hrybouski et al. thresholded and binarized the 3D NFT burden maps presented in our earlier work[32] to define tau-based MTL ROIs as seed for in vivo analysis of MTL-AT and MTL-PM functional connectivity. Such studies of both intra-MTL and MTL-dependent connectivity would benefit from the improved 3D mapping of NFT burden, and detailed, histology-based anatomical labeling of MTL subregions and entorhinal subfields presented in this work, to inform more accurate seed regions for connectivity analysis.

Our study has several potential limitations that are important to acknowledge. First, while postmortem MRI allows structure/pathology associations to be examined at a much greater resolution than possible with in vivo MRI, there are certain limitations associated with using cortical thickness measurements made from postmortem tissue. A study by Wisse et al.[57] compared cortical thickness of MTL substructures measured using in vivo (3T MRI), and ex vivo (9.4T MRI) scans of the same subject and found differences in thickness across the two scans. These differences were attributed to various factors such as 1) brain shrinkage caused by formalin fixation, 2) brain swelling caused by hypoxia and ischemia after or during death, and 3) increased brain size following extraction caused by a relief of intracranial pressure after autopsy. To the best of our knowledge, these factors are not linked to pathology. Despite these sources of variability, our analyses reveal patterns of correlation consistent with previous studies. However, further comparative studies are needed to help us better understand how to account for this potential source of variability in postmortem analyses[58]. Moreover, performing structure/pathology analyses using antemortem imaging has its own set of limitations, particularly when the time between antemortem imaging and death is long, since the postmortem pathology may not accurately reflect the state of pathology at the time of imaging. In future work, we will explore alternatives to cortical thickness as a measure of neurodegeneration, such as quantitative maps of cell density derived from serial histology, as a more direct measure of local neuronal injury due to NFT pathology.

While the machine learning classifier used to generate the NFT burden maps can reliably detect tangle-like inclusions, a second limitation of our algorithm is that it does not distinguish pre-tangles and

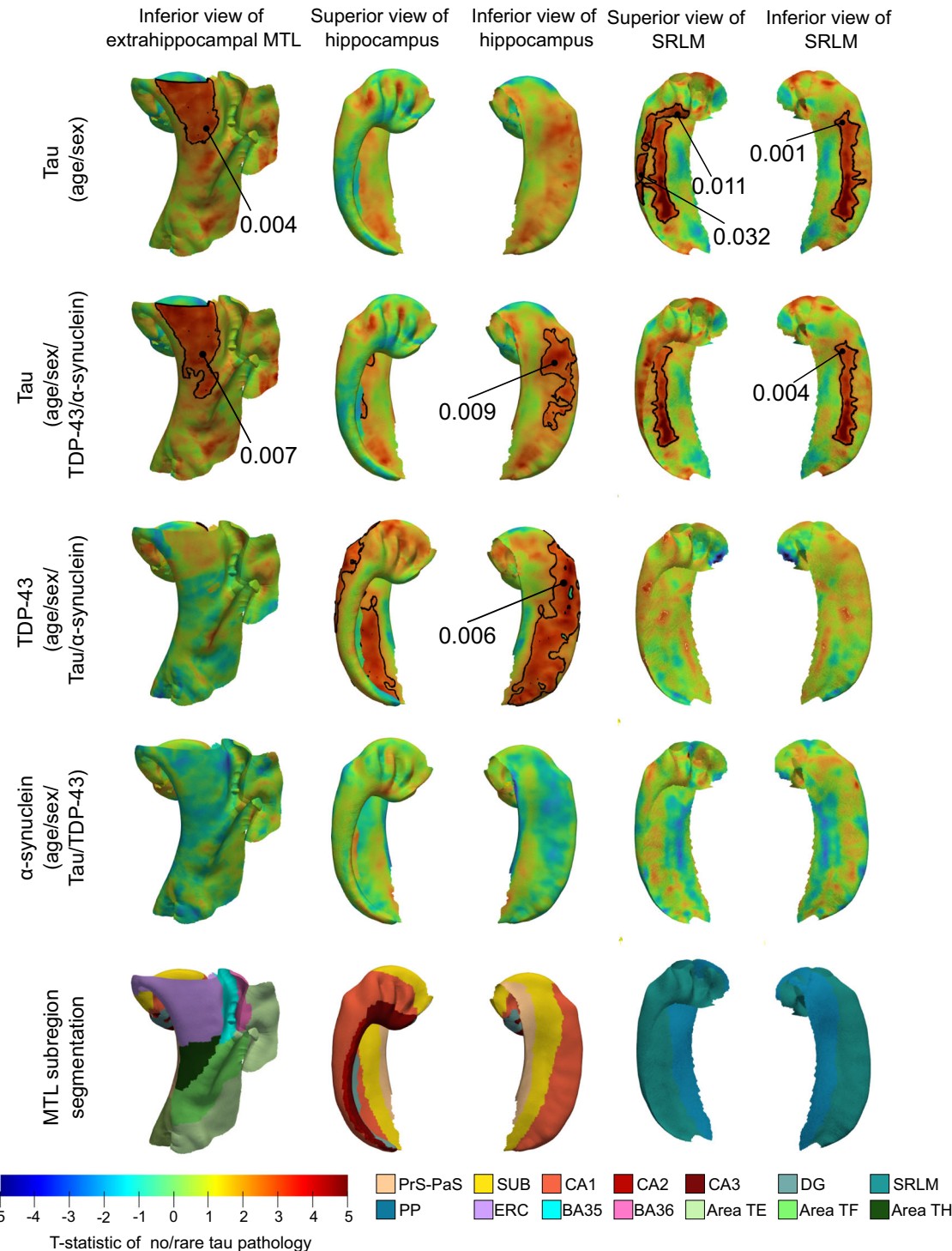

**Fig. 5 | Association between pointwise medial temporal lobe (MTL) thickness and contralateral, semi-quantitative neuropathology measures.** The t-statistic maps show the correlation between cortical thickness and the semi-quantitative neuropathology ratings based on tissue samples obtained from the MTL contralateral to the thickness measures ($n = 47$). Each row specifies the variable of interest with the covariates used in the analysis in parentheses. The first two rows show the association between thickness and tau burden, with and without correction for co-pathologies. Rows 3 and 4 show the patterns of correlation obtained between pointwise thickness and TDP-43 and $\alpha$-synuclein pathology respectively. Clusters were defined based on an empirical threshold (uncorrected $p < 0.01$) and

permutation testing with the Freedman & Lane method (1000 iterations) was used to assign each potential cluster a one-sided, corrected $p$-value. To account for multiple comparisons, the analysis uses cluster-level family-wise error rate correction. The clusters outlined in black indicate regions where a significant correlation was observed after correction for multiple hypothesis testing (corrected $p < 0.05$). No covariates were included in this model. (S subiculum, PrS-PaS: Pre/Parasubiculum, SRLM stratum radiatum lacunosum molecular, PP: perforant pathway, CA cornu ammonis, DG dentate gyrus, ERC entorhinal cortex, BA Brodmann area).

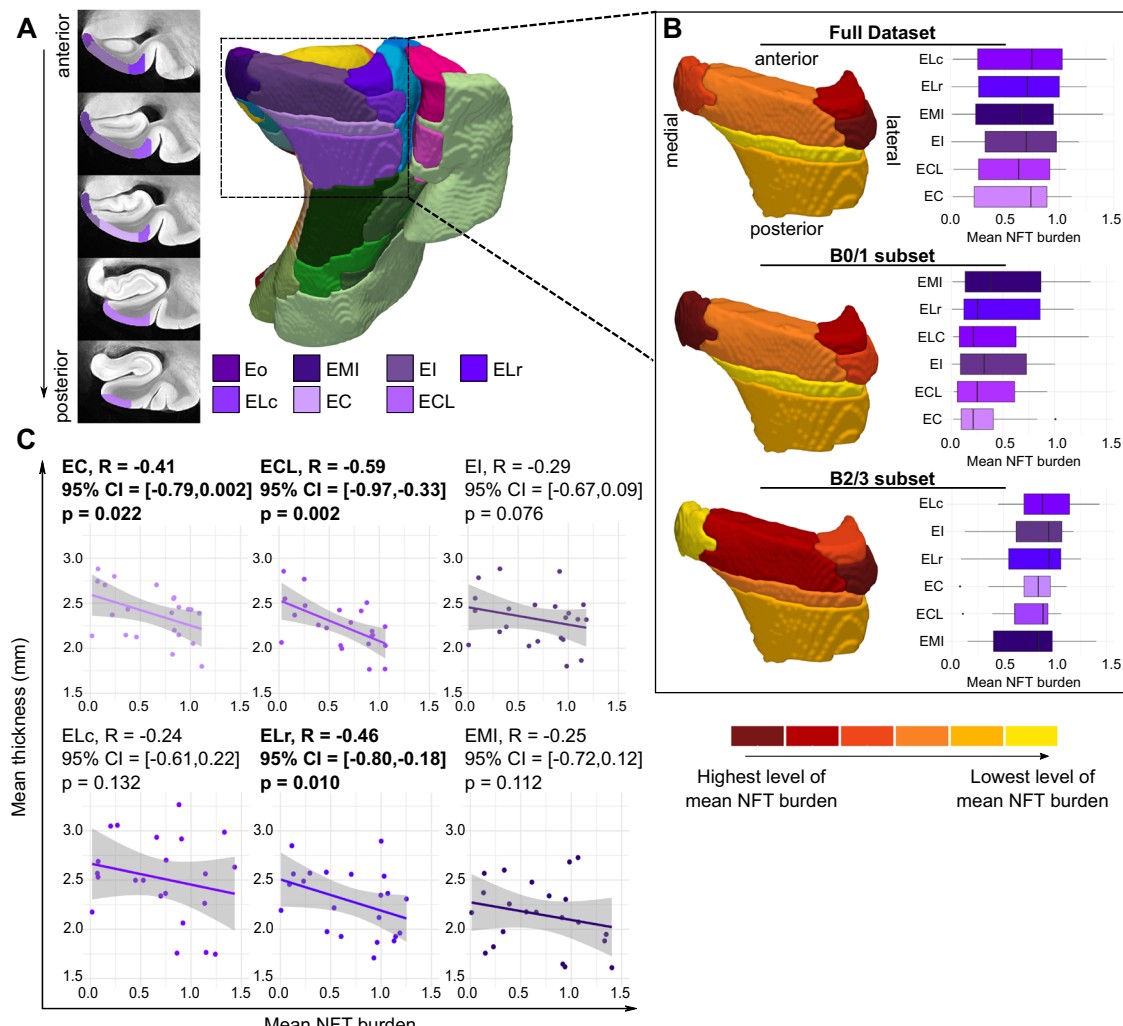

**Fig. 6 | 3-D distribution of neurofibrillary tangle pathology within the entorhinal cortex. A** Coronal cross-sectional view of the medial temporal lobe atlas at different levels, ordered from anterior to posterior, showing the histology-based consensus labeling of the entorhinal subfields. Note that since the anterior border of the atlas only starts at the hippocampal head, we are partially missing the anterior extents of the entorhinal subfields Eo, ER, and EMI. **B** Box and whisker plots showing the distribution of mean NFT burden across the different entorhinal subfields, computed using all cases and separately for cases with low (B0/1; $n = 11$) and high (B2/3; $n = 14$) B scores respectively. These plots are shown together with

corresponding surface heat maps which highlight the distribution of NFT burden across the entorhinal subfields ranked in order of highest to lowest mean NFT burden. **C** Scatter plots showing the relationship between mean cortical thickness and mean NFT burden computed within each subfield. Each plot also includes the Spearman's rank correlation. Significant negative associations are bolded (one-sided, uncorrected $p < 0.05$). Sample sizes are provided in Supplementary Table 3. Source data for 6(**B**) and (**C**) are provided as a Source Data file. (Eo: olfactory; EMI: medial intermediate; EI: intermediate; ELr: lateral rostral; ELc: lateral caudal; EC: caudal; ECL: caudal limiting).

astrocytic tau inclusions, which can be similar in visual appearance to tangles. To overcome these limitations, future work will focus on expanding the histological training dataset to include a wider variety of pathological inclusions and building on the frameworks developed thus far to generate 3D maps quantifying other forms of tau pathology (i.e., neuropil threads, neuritic plaques, astrocytic inclusions) as well as other neurodegenerative co-pathologies. We also aim to develop quantitative histopathological markers of neuroinflammation and vascular disease to be able to examine their role in neurodegenerative processes. Using these rich quantitative datasets, future analyses can be performed leveraging the atlas to better characterize the relationship between tau burden, co-morbid neurodegenerative pathologies, neuronal density, and cortical thinning.

In summary, in this study we generate an ex vivo MRI atlas with detailed cytoarchitecture-based labels at the level of MTL subfields, to describe, using quantitative histology measures, the 3-D distribution of NFT burden within the MTL and its regional effects on

neurodegeneration. Our findings provide a more refined postmortem reference for histological validation of future in vivo neuroimaging studies investigating tau spread in AD and have been uploaded to a publicly available data repository to facilitate their use in in vivo MRI and PET research. To improve our mapping of ex vivo information to in vivo structural MRI, we have been acquiring in vivo and ex vivo scans from the same subject that will enable more accurate matching of anatomical regions between the two domains. We have also been acquiring high-resolution, ex vivo scans of whole hemispheres and ongoing efforts are being made to develop tools and pipelines to examine tau/structure associations at the level of whole hemispheres[59,60]. Therefore, the current work also serves as a foundation for future ex vivo studies, leveraging larger datasets and more expansive quantitative histological characterizations, to examine associations between pathology and cognitive metrics, connectivity patterns, and other neuroimaging features, thus improving our understanding of AD biology and progression.

## Methods

### Ex vivo imaging procedure

Human brain specimens from CNDR and HNL were obtained in accordance with the local laws and regulations, and the Ethical Committee of UCLM respectively. Where possible, pre-consent during life and, in all cases, next-of-kin consent at death was given. For each brain donation, tissue from one hemisphere was used for imaging, and the opposite hemisphere was sampled for diagnostic pathology following the NIA/AA protocol[61,62]. Before imaging, CNDR hemispheres were fixed in a 10% formalin solution for at least 30 days. HNL hemispheres were initially fixed in situ by perfusion with 8 L of 4% paraformaldehyde through both carotid arteries and then stored until processing in a cold room, submerged in 4% paraformaldehyde[63]. After fixation, the temporal lobe was extracted from every hemisphere and imaged overnight on a Varian 9.4 T animal scanner at a 200 x 200 x 200 μm³ resolution using a multi-slice spin echo sequence. Sequence parameters vary slightly between specimens, with typical values being a repetition time of 9330 ms, and an echo time of 23 ms. Further details of the MRI acquisition and processing protocol are provided in Supplementary material, section 1.1.

Following MRI scanning, each MTL specimen underwent dense serial histological processing[63]. First, the specimens were cut into 2 cm blocks using custom molds that were 3-D printed to fit each temporal lobe specimen. The blocks were then cryoprotected and sectioned at 50 μm intervals in a sliding microtome coupled to a freezing unit (Microm, Heidelberg). Every 10th section was stained for Nissl using the 0.25% Thionin stain, resulting in approximately 40 sections per block. In addition, for a subset of 25 "AD continuum" cases, every 20th section was stained using AT8, a human phosphorylated tau antibody IHC stain, and counterstained for Nissl (~20 sections per block). Stained sections were then mounted on 75 mm x 50 mm glass slides, digitally scanned at 20X resolution, and uploaded to an open-source cloud-based digital histology archive (https://github.com/pyushkevich/histoannot) that supports web-based visualization, anatomical labeling, and machine learning classifier training. For each specimen, the scanned sections were reconstructed into a 3-D volume and aligned to 9.4 T MRI space using a custom deformable registration pipeline, described in[32]. The registration pipeline was evaluated in[32] by computing the distance between anatomical curves drawn independently on MRI and Nissl images, and overall, the registration accuracy was high (average distance < two MRI voxels for all curves).

### Construction of a computational ex vivo MRI atlas

To perform group analyses, we constructed a computational atlas of the MTL by applying a custom group-wise registration pipeline to the ex vivo MRI scans of 55 MTL specimens. The atlas construction pipeline described in[31], relies on shape information from segmentations of the MTL and SRLM in each MRI scan to guide the alignment of the MTL across different specimens. Twenty of the cases used in the current study were included in[31] and therefore already had MTL segmentations generated using a semi-automated interslice interpolation technique[64]. More recently, we have developed a more automated approach for combined MTL and SRLM segmentation based on a supervised convolutional neural network (CNN) framework that further reduces the time required to generate these segmentations. Our CNN is based on a modified implementation of the nnU-Net framework[65] that has been trained using a custom Laplacian-based loss function that we developed to improve the detection of sulci in the cortex[66]. One of the limitations of the previous iteration of the atlas was that only the more lateral portion of the SRLM layers was segmented, which limited our ability to accurately measure thickness in the full subiculum region. In the current work, we rectify this by training the developed CNN framework using an updated ground truth SRLM segmentation protocol in which the SRLM label includes the perforant pathway, extending over the entire subiculum and, pre- and parasubiculum. Additional details on the semi-automated approach used to update the ground truth SRLM segmentations, and CNN training and evaluation are provided in Supplementary material, sections 1.2 and[66]. This CNN framework was used to initialize MTL and SRLM segmentations for the remaining 35 cases in our dataset and to update the SRLM segmentations in the 20 cases previously used in[31]. For all cases, the predicted segmentations were visually checked, and any under-segmentations and errors in detecting the correct boundaries for the anterior and posterior MTL, and between the perirhinal and parahippocampal cortices were manually corrected. During this step, additional segmentation labels were added to distinguish the medial and lateral portions of the collateral sulcus, and to indicate regions of the scan affected by tearing or image artifacts (e.g., air bubbles), as was done in[31].

Using these segmentations, the group-wise volumetric image registration pipeline was applied as previously described, resulting in a synthetic template image that captures the "average" MTL anatomy across all 55 cases, the corresponding template segmentation, and a set of non-linear diffeomorphic transformations between the template and each specimen's scan. We labeled 26 MTL subregions, including subdivisions of the ERC, BA35, BA36, and Area TF, in 17 of the 55 specimens used to construct the atlas. Borders between subregions were first traced on serial Nissl digital histology images based on cytoarchitectural features. These borders were then mapped to 9.4T MRI space and used to trace 3-D segmentations of the subregions. Following atlas construction, the 17 completed histology-based MTL subregion segmentations were mapped to the space of the MRI atlas using the deformable transformations generated by the groupwise registration pipeline, and a consensus labeling of the MRI atlas was obtained by application of voxel-wise majority voting among the 17 segmentations with slight regularization by a Markov Random Field prior[40]. Details of the protocol used for cytoarchitectural annotation of the MTL subregions in each specimen are provided in[31], which followed a similar procedure but used 11 annotated specimens and fewer anatomical labels. We note that in the perirhinal cortex, the MTL subregion label Area TE refers to the portion of tissue lateral to BA36. We define the lateral border of Area TE as the medial bank of the occipitotemporal sulcus and separate the parahippocampal cortex (PHC) (consisting of Areas TF and TH) from Area TE based on the overall cytoarchitecture of the cortex, where TE is a typical neocortical region that corresponds to the association cortex, while PHC is called "proisocortex", that is, a structurally less complete (more primitive) type of cortex.

### Semi-quantitative neuropathology ratings

Semi-quantitative neuropathological evaluations are routinely performed by an expert neuropathologist at the CNDR to evaluate the severity of tau, Aβ, TDP-43, and α-synuclein pathology burden in a specimen using tissue sampled from 18 brain regions, following the NIAAA protocol[61]. In this study, we considered neuropathology ratings provided for three MTL regions examined in the CNDR neuropathology evaluations: the ERC, dentate gyrus (DG), and CA1/subiculum region[62]. At each region, the sampled tissue is embedded in paraffin blocks and cut into 6 μm sections for IHC using phosphorylated tau PHF-1 (mAb, 1:1000, a gift from Dr. Peter Davies) to detect phosphorylated tau deposits, NAB228 (monoclonal antibody [mAb], 1:8000, generated in the CNDR) to detect amyloid-β deposits, pS409/410 (mAb, 1:500, a gift from Dr. Manuela Neumann and Dr. E. Kremmer) to detect phosphorylated TDP-43 deposits and Syn303 (mAb, 1:16,000, generated in the CNDR) to detect the presence of the pathological conformation of α-synuclein. Each MTL location is visually assigned a semi-quantitative rating on a scale of 0-3 i.e., "none (0)", "rare (0.5)", "mild (1)", "moderate (2)" or "severe (3)"[61]. For our

analyses, we used the average rating across all three locations as a measure of pathology burden in the contralateral MTL.

## Quantitative 3-D maps of NFT burden derived from serial histological imaging

For the subset of 25 "AD continuum" cases with serial anti-tau IHC sections available, we generated 'heat maps' quantifying the burden of NFT pathology on individual anti-tau IHC sections using the machine learning algorithm described in Yushkevich et al. (2021)[32]. In brief, a weakly supervised deep learning algorithm was trained to classify patches of anti-tau IHC images labeled as containing either tangles (NFTs and pre-tangles) or non-tangles (tau neuropil threads, astroglial tau, tau coils in the white matter, normal tissue, slide background, artifacts, tissue folds) and achieved a test accuracy of >95%. During inference, in addition to assigning an input patch to a tangle versus non-tangle class, the network outputs a heat map, generated using the activation maps from the final layers of the network, indicating the location and intensity of any tangles in the image. This is illustrated in Supplementary Fig. 7. For each specimen, the generated IHC-derived NFT burden maps were co-registered to the space of ex vivo MRI and reconstructed in 3-D. The deformable registration pipeline used to align each tau IHC section to ex vivo MRI computes the transformations in two stages. First, each IHC section is aligned and deformed to the corresponding Nissl section and second, each Nissl section is aligned and deformed to the corresponding cross-section of the ex vivo MRI scan. The transformations for this second registration step are the same ones used to map the MTL subregion boundary annotations from Nissl histology to MRI space, as described in Section 2.3. To minimize the effects of any misregistrations on our downstream thickness analyses, before 3-D reconstruction, the results of the deformable registration pipeline were visually inspected on a per-section basis and sections with major misalignments in the MTL region were excluded from the final reconstruction. For each specimen, a 'mask' volume was created in MRI space that specifies where in the image space IHC-based measures are available, thus indicating where tau IHC sections were excluded and any small gaps between tissue blocks.

To perform group-level analysis, the 3-D NFT burden maps for all 25 specimens were brought into the space of the ex vivo MRI atlas using the transformations generated by the atlas construction pipeline. Maps of average NFT burden were computed and visualized in the space of the atlas. For quantitative comparisons, a single summary measure of NFT burden for each ROI in each specimen was obtained by computing the mean of the NFT burden map across all voxels in that ROI for which IHC-based measures were available (i.e., the 'masked' region).

## Group-level statistical analysis

To assess the relationship between MTL cortex tau burden and neurodegeneration, we examined the partial linear correlation between cortical thickness and both the semi-quantitative neuropathology ratings and the quantitative NFT burden summary measures. Regional thickness of the MTL cortex, hippocampal gray matter, and SRLM were estimated by applying Voronoi skeletonization to the native space segmentations of each specimen[67]. The thickness measurements of all the specimens were then mapped onto the skeleton of the MTL and SRLM templates for pointwise group comparison. We note that this is different from the approach followed in[31], where we performed regional thickness analyses along the boundary surfaces of the MTL and SRLM templates. We found that estimating thickness along the skeleton of the template increases robustness to registration errors between individual specimens and the template. More details on this are provided in Supplementary material, section 1.3.

In the larger dataset ($n = 47$), pointwise pathology-structure correlations were performed to test the effects of tau, TDP-43, and α-

synuclein pathologies on the MTL and SRLM surfaces. To test the effect of each pathology measure on pointwise thickness, we used the open-source tool 'meshglm' (https://github.com/pyushkevich/cmrep/) to fit a general linear model at each vertex on both the MTL and SRLM skeletal surfaces with the neuropathology measurement of interest as the independent variable, thickness as the dependent variable, and age and co-pathology measurements as nuisance covariates. Before statistical analysis, spatial smoothing (diffusion) was applied to the thickness data using a simple implementation of the heat equation (diffusion parameter, $T = 4$). The diffusion method was modified to avoid the propagation of missing thickness values. Additionally, the statistical computation was adapted to handle missing data by accounting for the variable number of observations and therefore degrees of freedom at each vertex. Only vertices where at least 25% of observations were valid were included in the analyses. To account for multiple comparisons, the analysis uses cluster-level family-wise error rate correction[33]. Potential clusters were defined based on an empirical threshold (uncorrected $p < 0.01$) and permutation testing with the Freedman & Lane method (1000 iterations) was used to assign each potential cluster a corrected $p$-value[68].

In the subset of cases with quantitative NFT burden maps ($n = 25$), we performed pointwise analysis correlating MTL and SRLM thickness with summary NFT burden measures derived from different anatomical subregions. No covariates were included in these models. In addition, we conducted ROI analyses assessing the local relationship between NFT burden and thickness by performing one-sided, partial Spearman correlation analyses using the R package "ppcor". 95% confidence intervals were determined based on bootstrapping with 1000 iterations using the R package "boot". ROI thickness measures were computed as the mean thickness within each anatomical subregion based on the consensus MTL labeling. ROI analyses were performed with and without age included in the model as a covariate and because of the relatively small size of this dataset, we did not correct for multiple comparisons. We note that case HNL21 (age 45 years) was excluded from all group-level analyses since this case is an outlier in terms of age, with the next youngest brain donor being 57 years.

## Reporting summary

Further information on research design is available in the Nature Portfolio Reporting Summary linked to this article.

## Data availability

The anonymized raw and processed data including the subject MRI scans, and subject-level and group-level histology-based segmentations, and quantitative NFT burden maps generated in this study have been deposited in the OpenNeuro database under accession code ds004767 [https://doi.org/10.18112/openneuro.ds004767.v1.0.0][69]. Detailed demographics, neuropathological diagnosis, and semi-quantitative neuropathology ratings for each donor are provided in the Supplementary Information file. The medial temporal lobe subregion-level quantitative tau burden measurements and mean thickness measurements used in this study are provided in the Source Data. Source data are provided with this paper.

## Code availability

The code repositories for atlas construction (https://github.com/sadhana-r/exvivo_tau_atlas_scripts, https://doi.org/10.5281/zenodo.11123538)[70] and 3D reconstruction of histology sections and registration to MRI (https://github.com/pyushkevich/tau_recon_scripts) are publicly available. All of the tools utilized in this work are open-source.

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

## Acknowledgements

We gratefully acknowledge the tissue donors and their families. We also thank all the staff at the National Disease Research Interchange brain bank, Center for Neurodegenerative Disease Research (University of Pennsylvania), and Human Neuroanatomy Laboratory (University of Castilla-La Mancha—UCLM) for performing the autopsies and making the tissue available for this project, as well as Sonia Herreros and personnel of the Department of Pathology, Albacete University Hospital. This work was supported in part by the National Institute of Health (Grants R01 AG056014 (P.A.Y), RF1 AG069474 (P.A.Y, D.A.W), P30 AG072979 (D.A.W), P01 AG066597 (D.J.I) and U19 AG062418 (E.B.L/ D.A.W), by MultiPark - A Strategic Research Area at Lund University (L.E.M.W) and a UCLM research grant to the Human Neuroanatomy Laboratory.

## Author contributions

S.R., P.A.Y., D.A.W., J.A.D., E.B.L., and S.R.D. conceptualized the study and developed the methods. S.R., A.D., S.L., E.C., N.S., L.E.M.W., D.A.W., R.I., L.X., D.J.I., E.B.L., M.D.T., K.P., J.L.R., T.S., G.M., M.M.I.O., M.M.A.J., M.M., M.P.M.R., S.C.S., J.C.D.G., C.R.P., R.In., and P.A.Y. performed research to acquire and process the data. S.R., A.D., S.L., N.S., and P.A.Y. analyzed the data. S.R., L.E.M.W., D.A.W., S.R.D., R.In., and P.A.Y. interpreted data and discussed results; S.R. drafted the initial manuscript. All authors revised the manuscript and provided critical feedback. P.A.Y. was the principal designer and coordinator of the study and supervised the collection, analysis, and interpretation of the study data.

## Competing interests

D.A.W. has served as a paid consultant to Eli Lilly, GE Healthcare, and Qynapse. He serves on a DSMB for Functional Neuromodulation and is a site investigator for a clinical trial sponsored by Biogen. All of this is outside of this work. L.X. received personal consulting fees from Galileo CDS, Inc., outside of this work, and has become an employee of Siemens Healthineers since May 2022. The current study was started during his employment at the University of Pennsylvania and is outside of his work at Siemens. S.R.D. received consultation fees from Rancho Biosciences and Nia Therapeutics, outside of this work. The remaining authors declare no competing interests.
