## [Peer Review File · Nature Communications]

Postmortem imaging reveals patterns of medial temporal lobe vulnerability to tau pathology in Alzheimer's diseaseREVIEWER COMMENTS

Reviewer #1 (Remarks to the Author):

Key results (question from editor: What are the noteworthy results?):

This work carried out a detailed analysis of associations between cortical thickness in the medial temporal lobe as quantified by post mortem MRI, and NFT burden as quantified from histology. In addition to scientific results, the work combines an incredibly broad set of methodological contributions (donor selection, clinical evaluation, tissue processing, post mortem MR imaging, group template construction, anatomical labeling, sectioning, histological imaging, tau detection and quantification, statistical analysis of associations). Each step is well executed and their combination is described in detail.

Significance (question from editor: -Will the work be of significance to the field and related fields? How does it compare to the established literature? If the work is not original, please provide relevant references.):

With several potential interventions for Alzheimer's disease becoming available, the search for well characterized biomarkers of the disease is of critical and timely importance. This work will be of significance to the field of Alzheimer's disease research, and of neuroimage analysis in general. The manuscript does a good job discussing related work, and discussing areas where the results are confirmatory or contradictory. This manuscript builds on earlier work by the same group, but provides sufficiently significant advancements.

Validity (question from editor: Does the work support the conclusions and claims, or is additional evidence needed?)

Overall the methods are well designed and the level of support for the reported association is strong. However, I have two comments that should be addressed.

First, the claims made in this work rest on cortical thickness measurements made in fixed post mortem samples. The authors should include a discussion of how these measurements relate to cortical thickness in living people. Are any distortions introduced? How reproducible are those distortions? etc. The authors should discuss any literature that addresses these questions, and/or describe any potential limitations to the interpretation of the results.

Second, the term "NFT burden" is introduced in the abstract and introduction, but not sufficiently

defined here or in the methods section. The authors should add a brief description of what this quantity is measuring, and any potential limitations to its interpretation (if any), in addition to referencing their earlier work that shows a more detailed analysis.

Data and methodology (questions from editor: Is the methodology sound? Does the work meet the expected standards in your field? - Is there enough detail provided in the methods for the work to be reproduced?)

The MR images presented here appear to be of high quality. The alignment of images for template construction, and for NFT burden heatmaps appear of high quality. The strategy discussed for removing any potential registration errors from statistical analysis is sufficient. There is enough detail provided in the manuscript and the associated references that the work could be (hypothetically) reproduced.

Please note that while I can vouch for image quality and computational analysis presented here, details of image acquisition and clinical determinations are outside the scope of my expertise.

Regarding the quality of presentation, there is a minor error in Figure 1. The caption states "BA35 (in shades of pink), BA36 (in shades of blue)", but this does not match the figure legend. I believe blue and pink have been reversed.

Furthermore, there are some acronyms that appear to be defined after they are used, not defined at all, or defined inconsistently. The authors should be sure to define Area TF, Area TE, Area TH. Entorhinal cortex is referred to as EC in some places and ERC in other places. With the large number of acronyms referring to anatomical regions, it would improve clarity and consistency if they could be defined in one place, such as a table (if the journal format allows).

One minor comment, the statement "We combined the presubiculum and parasubiculum into a single ROI, and the subdivisions within the entorhinal cortex, BA35 and BA36 respectively" was difficult for me to parse. I would suggest "We combined the presubiculum and parasubiculum into a single ROI, and did the same for the subdivisions within the entorhinal cortex, those within BA35 and those within BA36"

Analytical approach (question from the editor: Are there any flaws in the data analysis, interpretation and conclusions? - Do these prohibit publication or require revision?)

Other than the points raised above regarding potential distortions due to brain extraction and fixation, the analysis interpretations and conclusions appear sound. That being said, there is one statement about age as a potential confounder that should be explained in more detail:

"Significant correlations are observed across all subregions ($R > 0.53$ across all ROIs) suggesting that by co-varying for age in these analyses, we may be obscuring some of the associations due to NFT pathology."

It is not clear how this statement affects the interpretation of results, particularly in relation discussion points describing other datasets with different age distributions. It would be very helpful if the authors could expand.

Suggested improvements

I have no additional suggestions for improvements beyond what is discussed above.

Clarity and context

Results are presented clearly in the context of other work, but please note by above comment with respect to the definition of NFT burden. I believe this should be supplied in more detail in this work.

References

The references provided are adequate.

Please note the biorxiv reference has been published in Communications Engineering: <https://doi.org/10.1038/s44172-022-00044-1>, and the reference should be updated.

My area of expertise

As mentioned above, please note that while I can vouch for image quality and computational analysis presented here, details of image acquisition and clinical determinations are outside the scope of my expertise.

Reviewer #2 (Remarks to the Author):

The authors wrote a methodological manuscript on the MTL in AD. Specifically, they made quantitative NFT burden maps from serial anti-tau immunohistochemistry sections in 25 brain donors. Next, a detailed insight into the 3D-NFT spread across the human MTL at early vs. late stages of AD was shown. Additionally, they correlated MTL thickness with histological neuropathology measures and identified NFT-specific atrophy patterns. The authors motivate the use of measures from specific MTL subfields.

The work published in the manuscript is a true tour de force and fantastic addition to the field of neuroscience. With a strength in methodology, skill, and ingenuity, the manuscript also contains some textual weaknesses in regards to novelty and relevance.

The authors focus on NFT accumulation in Alzheimer's disease as their main objective (first sentence), however most of the manuscript (methods, results, discussion) is more about the methodology of the 3D NFT maps. I'd suggest to rephrase several paragraphs to focus more on the relevance of the methods and results for AD, how should the results be interpreted and implemented? And what is really the difference between the previous publication and the current one? Except for a larger N and more fine-grained visualization? How is this relevant to the AD (research) field? What additional insights does the current manuscript provide?

In the introduction they mention that their 3D approach would give a more complete picture than 2D approaches, but I don't see this shown in the results or sufficiently discussed (with examples) in the discussion. Furthermore, they mention in the abstract and introduction that their insights could help inform imaging biomarkers by defining regions of the MTL where neurodegeneration is most directly linked to underlying NFT pathology opposed to other comorbidities. In the results and discussion I miss a direct answer to this and the novelty/addition to what's already known (e.g. subiculum and CA1). Now it seems like they are confirming the current (e.g. Braak) literature in a fancy way, but I'm sure there is so much more the NFT maps could add.

Coming back to the focus on NFT in AD, it strikes me that there wasn't a more stringent selection of cases included in the study. From table 1 and supplementary table 1, I notice some LBD cases with comorbid AD pathology. However, for correlations with e.g. thickness, the primary pathology may play a role too. Furthermore, PART may be tau related, but has a different distribution than the Braak stages. In which way is this important to the NFT maps? Or for NFT maps in AD specifically?

In the subject tables I miss information on clinical diagnosis, cause of death, and postmortem interval. Also, statistical corrections were made for age (although not very consistently), but no correction for sex or (intracranial) volume. Now I can imagine the latter is difficult when only the MTL is taken out, but should then be mentioned as a limitation. Actually, the limitation section is missing in the discussion.

Some additional minor points:

- the discussion is quite lengthy and can be written in a more concise manner.
- semi-quantitative data of the contralateral side seems superfluous when you have quantitative data

from the ipsilateral side, I'd suggest to move paragraph 2.3.1 to the supplement, and not have it in the results/discussion as much. Focus on the quantitative data.

- Figure 4 means an explanation for PP.
- the abstract is very methodological, what are the main results and conclusion?
- I'm missing a (figure) example of how quantification is done/measured?
- Figure 2 "frequency of NFT burden" and "average NFT burden" have overlapping color schemes in the same image?
- Figure 6 "ranking in order of decreasing mean NFT burden level" is 1-6, which is similar to increasing NFT Braak staging, confusing?

Reviewer #3 (Remarks to the Author):

Ravikumar et al. investigated the pattern of tau neurofibrillary tangles (NFTs) and its association with cortical thickness in Alzheimer's disease (AD) post-mortem brain tissues using a novel 3-D quantitative map of histological analysis of AD pathology. In this study, the authors included 55 donors aged between 57-99 years to support the overall stereotypical spreading of Braak staging although there were some diverging regions that may have been difficult to evaluate appropriately in the traditional 2-D slices of the brain tissues. Moreover, the study shows association between the NFTs and other co-pathology such as TDP-43 and alpha-synuclein to be associated with cortical thickness. The study methodology and design are elegant, and I commend all the work that went into these histological studies, ex-vivo scanning, and the mapping between the different modalities. And although the sample size may seem small for an in-vivo study, this is quite a large sample size for such ex-vivo studies. While I appreciate the advanced mapping & registration methods, the use/creation of the ex-vivo atlas, and inclusion of other co-pathologies to provide a fine-grained, comprehensive mapping of tau pathology, several issues and questions need to be addressed or further clarified.

Major comments:

- The authors could better describe the new findings of this current paper compared to their prior recent publication (Yushkevich et al. Brain 2021). As presented, the current study seems largely an extension of their prior work with a bigger sample size, early vs. late stages along the AD continuum, and some improvements on the atlas (e.g. full extent of hippocampal SRLM layer). For example, I find the sentence in the abstract "characterize, for the first time, three-dimensional NFT spread across the human MTL at the early vs. late stages of AD" somewhat misleading as the authors already have a publication on their 3D quantification technique in the MTL. Similarly, in the discussion: "we characterize at an unprecedented level of detail, the 3-D probabilistic distribution of NFT burden at the different stages of AD". however, what is unprecedented is (the application of their technique in) a larger cohort that encompasses the AD continuum. This paper is not describing a "novel" technique. What is new is the creation of an ex-vivo MR atlas consisting of cytoarchitecture-guided segmentations of MTL subregions, and it's not entirely clear what new insights the improved atlas and additional data are providing.

- The current AD literature supports that the distribution of tau pathology is strongly associated with the symptoms and/or phenotypes observed in AD patients. Given that the study showed the evidence that the brain tissues investigated followed the typical spreading of tau described by Braak and Braak, is possible for the authors to provide any cognitive or behavioral metrics and associate them with the pathology?
- Based on the description, it seems like the authors reported the main effect of NFTs, TDP-43, or a-synuclein on neurodegeneration. Were there any significant interactive effects between the different co-pathologies?
- Linking NFT patterns to thickness on this high-res 3D level is fascinating but remains confined to the MTL. How do the authors see an expansion/translation of their work to larger parts of the brain? As in-vivo literature has shown neocortical tau spread along neural connections in the brain (both functionally and structurally), how could different [connected] parts of the brain be integrated within a more continuous 3D model of tau spread?
- Is there a mapping between the original in-vivo atlas of the prior work and the new ex-vivo atlas? It would be quite interesting to see if the NFT burden trends still hold and how they compare in the space of a (down-sampled) in-vivo atlas with (artificially) much larger voxel sizes compared to the ex-vivo atlas - mimicking an in-vivo structural MRI or PET scan.
- Was quantitative immunohistochemistry performed in some of the tissue samples for any of the pathologies? And how did it correspond with the semi-quantitative ratings?

Minor comments:

- Given the possible histological examinations, have the authors investigated any neuroinflammatory or vascular biomarkers/cells?
- Is there any information on APOE status of the subjects?
- In exploratory and uncorrected analyses (for multiple corrections), it would be beneficial to perform some sort of bootstrapping or permutations and provide ranges of correlations and effect sizes or p-values.
- Numerous neuroimaging studies suggest that the structural and/or functional connectomes may be implicated in tauopathy and its spreading. As the proposed method combine histological data with neuroimaging space, can the authors discuss how such structural and/or functional networks can be addressed?
- SRLM should be defined in the abstract.
- Figure 4: the pvalue of the last panel (area TH) seems incorrect.
- Table 1: what does the primary vs secondary diagnoses refer to?

AUTHORS RESPONSE TO REVIEWERS

RE: We thank the reviewers for their constructive comments and helpful feedback. We have adapted our manuscript based on their suggestions and recommendations. Please see below for a point-by-point response to the reviewer's comments and concerns, with the relevant sections of the manuscript included. Changes to the main manuscript are shown in red colored text.

Reviewer #1 (Remarks to the Author):

Key results (question from editor: What are the noteworthy results?):

This work carried out a detailed analysis of associations between cortical thickness in the medial temporal lobe as quantified by post mortem MRI, and NFT burden as quantified from histology. In addition to scientific results, the work combines an incredibly broad set of methodological contributions (donor selection, clinical evaluation, tissue processing, post mortem MR imaging, group template construction, anatomical labeling, sectioning, histological imaging, tau detection and quantification, statistical analysis of associations). Each step is well executed and their combination is described in detail.

Significance (question from editor: -Will the work be of significance to the field and related fields? How does it compare to the established literature? If the work is not original, please provide relevant references.):

With several potential interventions for Alzheimer's disease becoming available, the search for well characterized biomarkers of the disease is of critical and timely importance. This work will be of significance to the field of Alzheimer's disease research, and of neuroimage analysis in general. The manuscript does a good job discussing related work, and discussing areas where the results are confirmatory or contradictory. This manuscript builds on earlier work by the same group, but provides sufficiently significant advancements.

RE: Thank you!

Validity (question from editor: Does the work support the conclusions and claims, or is additional evidence needed?)

Overall the methods are well designed and the level of support for the reported association is strong. However, I have two comments that should be addressed.

1) First, the claims made in this work rest on cortical thickness measurements made in fixed post mortem samples. The authors should include a discussion of how these measurements relate to cortical thickness in living people. Are any distortions introduced? How reproducible are those distortions? etc. The authors should discuss

any literature that addresses these questions, and/or describe any potential limitations to the interpretation of the results.

RE: As suggested, we now include in the discussion section the potential limitations associated with cortical thickness measurements made using postmortem MRI:

Discussion pg. 16: “Our study has several potential limitations that are important to acknowledge. First, while postmortem MRI allows structure/pathology associations to be examined at a much greater resolution than possible with in vivo MRI, there are certain limitations associated with using cortical thickness measurements made from postmortem tissue. A study by Wisse et al. [56] compared cortical thickness of MTL substructures measured using in vivo (3T MRI), and ex vivo (9.4T MRI) scans of the same subject and found differences in thickness across the two scans. These differences were attributed to various factors such as 1) brain shrinkage caused by formalin fixation, 2) brain swelling caused by hypoxia and ischemia after or during death and 3) increased brain size following extraction caused by a relief of intracranial pressure after autopsy. To the best of our knowledge, these factors are not linked to pathology. Despite these sources of variability our analyses reveal patterns of correlation consistent with previous studies. However, further comparative studies are needed to help us better understand how to account for this potential source of variability in postmortem analyses [57]. Moreover, performing structure/pathology analyses using antemortem imaging has its own set of limitations, particularly when the time between antemortem imaging and death is long, since the postmortem pathology may not accurately reflect the state of pathology at the time of imaging. In future work, we will explore alternatives to cortical thickness as a measure of neurodegeneration, such as quantitative maps of cell density derived from serial histology, as a more direct measure of local neuronal injury due to NFT pathology.”

2) Second, the term "NFT burden" is introduced in the abstract and introduction, but not sufficiently defined here or in the methods section. The authors should add a brief description of what this quantity is measuring, and any potential limitations to its interpretation (if any), in addition to referencing their earlier work that shows a more detailed analysis.

RE: We now introduce the use of a machine-learning based approach for computing NFT burden directly in the introduction. We also include a more detailed description of how NFT burden is measured in the methods section and a supplementary figure illustrating the weakly supervised segmentation pipeline used to derive 3D tau tangle burden maps (see Supplementary Fig. 7), in addition to referencing earlier work that provides more detailed methods. We have also added to the discussion section the potential limitations associated with our automated NFT burden measures.

Introduction pg. 2: “In this study we leverage foundational frameworks presented in our earlier work to construct a computational atlas of the MTL using ex vivo MRI, which enables statistical mapping of associations between MTL cortical/hippocampal thickness and multiple markers of pathology [31], and generate 3-D quantitative “heat

maps" of NFT pathology from dense serial histology using machine learning algorithms [32]. Yushkevich et al. [32] showed that NFT burden measures derived from these heat maps (i.e. the mean intensity across the heat map) correlate strongly with manual tangle counts and semi-quantitative ratings of NFT severity provided by an expert neuropathologist, indicating that they accurately capture both the number and severity of tangle-like pathological inclusions in the tissue."

Methods pg. 18: "For the subset of 25 "AD continuum" cases with serial anti-tau IHC sections available, we generated 'heat maps' quantifying the burden of NFT pathology on individual anti-tau IHC sections using the machine learning algorithm described in Yushkevich et al. (2021) [32]. In brief, a weakly supervised deep learning algorithm was trained to classify patches of anti-tau IHC images labelled as containing either tangles (NFTs and pre-tangles) or non-tangles (tau neuropil threads, astroglial tau, tau coils in the white matter, normal tissue, slide background, artifacts, tissue folds) and achieved a test accuracy of >95%. During inference, in addition to assigning an input patch to a tangle versus non-tangle class, the network outputs a heat map, generated using the activation maps from the final layers of the network, indicating the location and intensity of any tangles in the image. This is illustrated in Supplemental Fig. 7."

Discussion pg. 16: "While the machine learning classifier used to generate the NFT burden maps is able to reliably detect tangle-like inclusions, a second limitation of our algorithm is that it does not distinguish pre-tangles and astrocytic tau inclusions, which can be similar in visual appearance to tangles. To overcome these limitations, future work will focus on expanding the histological training dataset to include a wider variety of pathological inclusions and building on the frameworks developed thus far to generate 3D maps quantifying other forms of tau pathology (i.e. neuropil threads, neuritic plaques, astrocytic inclusions) as well as other neurodegenerative co-pathologies. We also aim to develop quantitative histopathological markers of neuroinflammation and vascular disease to be able to examine their role in neurodegenerative processes. Using these rich quantitative datasets, future analyses can be performed leveraging the atlas to better characterize the relationship between tau burden, co-morbid neurodegenerative pathologies, neuronal density, and cortical thinning."

Data and methodology (questions from editor: Is the methodology sound? Does the work meet the expected standards in your field? - Is there enough detail provided in the methods for the work to be reproduced?)

The MR images presented here appear to be of high quality. The alignment of images for template construction, and for NFT burden heatmaps appear of high quality. The strategy discussed for removing any potential registration errors from statistical analysis is sufficient. There is enough detail provided in the manuscript and the associated references that the work could be (hypothetically) reproduced.

Please note that while I can vouch for image quality and computational analysis presented here, details of image acquisition and clinical determinations are outside the

scope of my expertise.

3) Regarding the quality of presentation, there is a minor error in Figure 1. The caption states "BA35 (in shades of pink), BA36 (in shades of blue)", but this does not match the figure legend. I believe blue and pink have been reversed.

RE: We have corrected the color references in the figure legend.

4) Furthermore, there are some acronyms that appear to be defined after they are used, not defined at all, or defined inconsistently. The authors should be sure to define Area TF, Area TE, Area TH. Entorhinal cortex is referred to as EC in some places and ERC in other places. With the large number of acronyms referring to anatomical regions, it would improve clarity and consistency if they could be defined in one place, such as a table (if the journal format allows).

RE: We have edited the manuscript to refer to the entorhinal cortex consistently as ERC in the text and figures. We now also include a table that lists abbreviations and descriptions for the different anatomical labels included within the medial temporal lobe segmentation, including Area TE, TF, and TH (see Table 2). We have also added to the Methods section, a more detailed definition of Area TE and the parahippocampal cortex (Areas TF and TH).

Methods pg. 18: "We define the lateral border of Area TE as the medial bank of the occipitotemporal sulcus and separate the parahippocampal cortex (PHC) (consisting of Areas TF and TH) from Area TE based on the overall cytoarchitecture of the cortex, where TE is a typical neocortical region that corresponds to association cortex, while PHC is called "proisocortex", that is, a structurally less complete (more primitive) type of cortex."

5) One minor comment, the statement "We combined the presubiculum and parasubiculum into a single ROI, and the subdivisions within the entorhinal cortex, BA35 and BA36 respectively" was difficult for me to parse. I would suggest "We combined the presubiculum and parasubiculum into a single ROI, and did the same for the subdivisions within the entorhinal cortex, those within BA35 and those within BA36"

RE: As suggested, we have rephrased the sentence.

Analytical approach (question from the editor: Are there any flaws in the data analysis, interpretation and conclusions? - Do these prohibit publication or require revision?)

6) Other than the points raised above regarding potential distortions due to brain extraction and fixation, the analysis interpretations and conclusions appear sound. That

being said, there is one statement about age as a potential confounder that should be explained in more detail:

"Significant correlations are observed across all subregions ($R > 0.53$ across all ROIs) suggesting that by co-varying for age in these analyses, we may be obscuring some of the associations due to NFT pathology."

It is not clear how this statement affects the interpretation of results, particularly in relation discussion points describing other datasets with different age distributions. It would be very helpful if the authors could expand.

RE: In the revised manuscript, we have now tried to clarify the referenced statement in the results and also include an expanded discussion of the potential effects of age in our analyses using the quantitative NFT burden maps. We have also included a new Supplemental figure showing the results of the pointwise analysis correlating thickness and quantitative NFT burden measures with age and sex as additional co-variates in the model (see Supplementary Fig. 4)

Results pg. 7: "Only NFT-structure associations in dentate gyrus, Area TE and Area TH remained significant when age was added to the model and no significant associations were observed when both age and sex were added to the model, likely due to the small sample size. The scatter plots show high levels of variability in cortical thickness measurements across specimens even at low levels of NFT burden, suggesting that non-AD developmental or aging-related variations in thickness across cases may be weakening the detected associations. In Supplementary Fig. 3, we include scatter plots showing the relationship between age and mean NFT burden computed within each of the anatomical ROIs. Significant correlations between age and NFT burden are observed across all subregions ($R > 0.53$ across all ROIs). Since age is likely in the causal pathway between the accumulation of NFT burden and neurodegeneration, with increased age leading to higher levels of tau pathology, which in turn leads to reduced cortical thickness, by correcting for age, we may be obscuring a crucial aspect of the pathway that we are interested in investigating, rather than correcting for a confounder."

Discussion pg. 15: "Indeed, the patient population used in the present study includes many older participants which could explain the high anterior tau burden observed even in the cases with low B scores (B0/1 subset: age range 59-93 years). While this is a relatively small sample size to study the effects of multiple covariates, this finding together with our result showing weakened tau-thickness associations in the ERC when accounting for age and sex effects (see Supplementary Fig. 4), highlight the importance of future studies examining age-related spreading of tau pathology, particularly within the ERC. The patterns of tau-thickness correlations observed in the current analyses using semi-quantitative tau ratings, which includes age and sex in the model, are consistent with the atrophy patterns detected in the analyses using quantitative NFT burden measures, suggesting that these neurodegenerative effects are largely driven by disease severity and not age-related factors. Furthermore, even though we observe a strong association between age and NFT burden in our dataset, we also observe

significant increases in NFT burden across all MTL subregions when comparing cases at low and high Braak stages. In the present study, we don't have a large enough sample size to investigate age-related pathology patterns and analyze the primary AD cases separately from other groups such as PART and those where AD is the secondary diagnosis. Our findings motivate future analyses using a larger dataset to perform tau mediation analyses and examine potential differences in 3D tau distributions and pathology-structure associations between diagnostic groups and age groups.”

Suggested improvements

I have no additional suggestions for improvements beyond what is discussed above.

Clarity and context

Results are presented clearly in the context of other work, but please note by above comment with respect to the definition of NFT burden. I believe this should be supplied in more detail in this work.

References

The references provided are adequate.

7) Please note the biorxiv reference has been published in Communications Engineering: <https://doi.org/10.1038/s44172-022-00044-1>, and the reference should be updated.

RE: Thank you for pointing this out. The reference has been updated.

My area of expertise

As mentioned above, please note that while I can vouch for image quality and computational analysis presented here, details of image acquisition and clinical determinations are outside the scope of my expertise.

Reviewer #2 (Remarks to the Author):

The authors wrote a methodological manuscript on the MTL in AD. Specifically, they made quantitative NFT burden maps from serial anti-tau immunohistochemistry sections in 25 brain donors. Next, a detailed insight into the 3D-NFT spread across the human MTL at early vs. late stages of AD was shown. Additionally, they correlated MTL thickness with histological neuropathology measures and identified NFT-specific atrophy patterns. The authors motivate the use of measures from specific MTL subfields.

The work published in the manuscript is a true tour de force and fantastic addition to the field of neuroscience. With a strength in methodology, skill, and ingenuity, the manuscript also contains some textual weaknesses in regards to novelty and relevance.

RE: We thank the reviewer for these supportive comments! We have revised the manuscript to better highlight the novelty and potential impact of this work on the field of AD research.

1) The authors focus on NFT accumulation in Alzheimer's disease as their main objective (first sentence), however most of the manuscript (methods, results, discussion) is more about the methodology of the 3D NFT maps. I'd suggest to rephrase several paragraphs to focus more on the relevance of the methods and results for AD, how should the results be interpreted and implemented? And what is really the difference between the previous publication and the current one? Except for a larger N and more fine-grained visualization? How is this relevant to the AD (research) field? What additional insights does the current manuscript provide?

RE: Following the reviewer's comment, we have reorganized and rephrased parts of the introduction and several paragraphs in the discussion to better describe the broader impact of our work and the potential utility of our findings towards the development of improved biomarkers and future in vivo studies investigating the spread of tau pathology in early AD. We discuss how the cytoarchitecture-based segmentations and regions of high NFT burden identified in our work can be used to define histologically validated MTL regions of interest as seeds for in vivo structural and functional connectivity analysis. We also highlight how future work can leverage the methodological frameworks used in this study to examine a broader range of neuropathologies and other brain regions of interest beyond the MTL.

We have also now added a supplementary figure comparing the 3D NFT maps presented in the current paper in comparison to the one presented in our previous publication (Yushkevich et al. (2021), both in the space of an in vivo template to highlight any differences and improvements achieved using the current approach. In Yushkevich et al. (2021), a coarse ex vivo atlas was first constructed by applying the conventional, image-based population template building algorithm to our ex vivo scans. This rough ex vivo template was then registered to the MNI in vivo atlas for visualization of the 3D NFT burden maps. Furthermore, the anatomical regions of interest (ROIs) used for describing anatomical differences in NFT distribution were based on in vivo

atlases that may not have matched individual anatomical structures well in the in vivo template space. Here, we use advanced, shape-based registration techniques to more accurately align the ex vivo scans and describe anatomical differences in NFT distribution using histology-based anatomical ROIs.

We would also like to highlight that unlike our previous studies, the current study includes novel analyses linking quantitative NFT burden measures with thickness measures obtained in the same region, and we limited our analyses to cases along the AD continuum. Taken together, these various methodological improvements and the better selection of brain donors result in a far more precise and reliable characterization of NFT pathology and its neurodegenerative effects within the MTL.

2) In the introduction they mention that their 3D approach would give a more complete picture than 2D approaches, but I don't see this shown in the results or sufficiently discussed (with examples) in the discussion. Furthermore, they mention in the abstract and introduction that their insights could help inform imaging biomarkers by defining regions of the MTL where neurodegeneration is most directly linked to underlying NFT pathology opposed to other comorbidities. In the results and discussion, I miss a direct answer to this and the novelty/addition to what's already known (e.g. sibiculum and CA1). Now it seems like they are confirming the current (e.g. Braak) literature in a fancy way, but I'm sure there is so much more the NFT maps could add.

RE: We have revised the results and discussion sections to better describe the insights gained from 3D characterization of tau pathology within the MTL and the analyses linking tau pathology and cortical thickness. We now also include a supplementary video showing 3D volumetric renderings of the average NFT burden maps computed separately for low and high Braak cases.

Results pg. 6: "In 2-D histological studies, sectioning is typically done in the coronal plane, thus limiting our knowledge on the distribution of NFTs along the anterior-posterior axis of the MTL. The sagittal views of our 3-D mappings reveal a marked anterior to posterior gradient in NFT burden along the parahippocampal gyrus, visible in cases at both the early and late Braak stages. We also observe increased NFT burden in CA1 along the full length of the hippocampus. These patterns are clearly depicted in 3-D visualizations comparing the average NFT burden maps at the early and late Braak stages, as shown in Supplementary Video 1."

Discussion pg. 11: "Leveraging postmortem imaging of a relatively large number of human MTL specimens with neuropathological diagnoses on the AD continuum, we characterize at an unprecedented level of detail, the 3-D probabilistic distribution of NFT burden at the different stages of AD and the regional effects of tau pathology on MTL neurodegeneration. This allows us to visualize and analyze patterns of NFT distribution along both the coronal and sagittal axes of the MTL, and thus offers more extensive

information than current histology-based descriptions of NFT topography in AD, which are in 2-D and based on selective sampling of the MTL [6,8,9,11].”

Discussion pg. 11: “Here, in addition to early transentorhinal involvement, we observe similar and in some cases higher levels of NFT burden in the CA1 subfield of the hippocampus, suggesting a more widespread distribution of NFT pathology during this early stage. We also observe greater vulnerability to NFT pathology in the anterior portion of the parahippocampal gyrus extending towards the temporal pole. This increased anterior involvement includes the region that appears to correspond to the amygdala, consistent with findings reported in previous neuropathology studies which have shown changes in the amygdala due to the presence of NFT pathology [8,10,36]. Overall, these findings add important histological evidence showing the broader impact of NFT pathology during early AD beyond just the transentorhinal region, highlighting the need for examining the hippocampal subfields and amygdala in future in vivo studies of early AD.”

Discussion pg. 14: “Together, the results of our regional thickness analyses suggest that the ERC and SRLM might be more specific to tau pathology, as opposed to aging and non-AD pathologies. We hypothesize that by mapping the detected ‘hotspots’ to the in vivo domain, in vivo measures of structural change derived from these regions would provide heightened sensitivity and utility over conventional whole hippocampus imaging biomarkers.”

4) Coming back to the focus on NFT in AD, it strikes me that there wasn't a more stringent selection of cases included in the study. From table 1 and supplementary table 1, I notice some LBD cases with co-morbid AD pathology. However, for correlations with e.g. thickness, the primary pathology may play a role too. Furthermore, PART may be tau related, but has a different distribution than the Braak stages. In which way is this important to the NFT maps? Or for NFT maps in AD specifically?

RE: We now include in the manuscript an explanation of how the primary versus secondary pathological diagnosis is determined:

Results pg. 3: “Note that in brains with co-occurring pathologies, the neuropathology that is more dominant or advanced is listed as primary neuropathological diagnosis. However, there are cases where it is not clear which neuropathology is more dominant. In such cases, the driving neuropathology that is more responsible for the clinical phenotype is listed as the primary neuropathological diagnosis even though both the “primary” and “secondary” pathological diagnoses are equally severe.”

With regards to the cases with Lewy Body Disease (LBD), we did not exclude LBD cases from our study since previous studies have indicated that α -synuclein pathology, the main component of Lewy Bodies, does not have a strong effect on MTL neurodegeneration. Indeed, in our analyses linking cortical thickness and semi-

quantitative ratings of neuropathology in the larger dataset, we do not observe any significant associations between α -synuclein pathology and cortical thinning across the MTL surface (see Figure 5). Furthermore, the observed significant associations between tau pathology and cortical thinning survive correcting for both co-morbid TDP-43 and α -synuclein as nuisance covariates in the model. Even though the analyses using quantitative NFT burden measures do not account for co-morbid α -synuclein pathology, our findings using contralateral, semi-quantitative data suggest that the neurodegenerative effects observed in this smaller dataset are most likely driven by NFT pathology. While there is some evidence that the pattern of NFT distribution might be different for LBD with secondary AD [1], current literature on this topic is sparse.

With regards to the cases with PART, PART is defined by the presence of NFTs, but with minimal or no amyloid-beta deposition. According to Crary et al (2014) [2], the distribution of NFT burden in PART is predominantly in the hippocampus and entorhinal cortex, consistent with the early Braak regions. Das et al. also found correlation patterns between tau burden and atrophy consistent with the early Braak stages in a PET study examining an amyloid-negative patient group [3]. While there might be differences in the distribution of NFTs in PART, with studies reporting increased levels of NFTs in CA2 in PART compared to AD [4,5], these differences are marginal and were not found in Zhang et al [6]. Therefore, since PART is generally thought to follow Braak stages with some evidence of subtle differences, it is unlikely to have a significant impact on our analyses.

Nevertheless, we agree that exploring difference in the distribution of NFT between different groups would be an interesting avenue for future research and now note this in the discussion section.

Discussion pg. 15: "In the present study, we don't have a large enough sample size to investigate age-related pathology patterns and analyze the primary AD cases separately from other groups such as PART and those where AD is the secondary diagnosis. Our findings motivate future analyses using a larger dataset to perform tau mediation analyses and examine potential differences in 3D tau distributions and pathology-structure associations between diagnostic groups and age groups."

[1] Coughlin, David, et al. "Cognitive and pathological influences of tau pathology in Lewy body disorders." *Annals of neurology* 85.2 (2019): 259-271.

[2] Crary, John F., et al. "Primary age-related tauopathy (PART): a common pathology associated with human aging." *Acta neuropathologica* 128 (2014): 755-766.

[3] Das, Sandhitsu R., et al. "In vivo measures of tau burden are associated with atrophy in early Braak stage medial temporal lobe regions in amyloid-negative individuals." *Alzheimer's & Dementia* 15.10 (2019): 1286-1295.

[4] Jellinger, Kurt A. "Different patterns of hippocampal tau pathology in Alzheimer's disease and PART." *Acta Neuropathologica* 136.5 (2018): 811-813.

[5] Walker, Jamie M., et al. "Early selective vulnerability of the CA2 hippocampal subfield in primary age-related tauopathy." *Journal of Neuropathology & Experimental Neurology* 80.2 (2021): 102-111.

[6] Zhang, Lei, et al. "Quantitative assessment of hippocampal tau pathology in AD and PART." *Journal of Molecular Neuroscience* 70 (2020): 1808-1811.

5) In the subject tables I miss information on clinical diagnosis, cause of death, and postmortem interval. Also, statistical corrections were made for age (although not very consistently), but no correction for sex or (intracranial) volume. Now I can imagine the latter is difficult when only the MTL is taken out, but should then be mentioned as a limitation. Actually, the limitation section is missing in the discussion.

RE: We have now added postmortem interval information to the extended demographics table (Supplementary Table 1) and included a more distinct limitations section in the discussion. We do not report cause of death and clinical diagnosis as we do not have this information consistently available for all the cases. However, we report neuropathological diagnosis which we believe is more relevant to the present study since it is biologically defined.

We have updated the results shown in Fig. 5, presenting the pointwise correlations between thickness and semi-quantitative neuropathological measures, to now include sex as an additional covariate.

We also repeated the thickness analyses performed using the quantitative NFT burden measures with sex included as a covariate in addition to age. Although it is important to note that the small sample size in these analyses ($n = 25$) limits our ability to reliably correct for multiple variables. All the MTL subregions show weakened associations between regional NFT burden and thickness when both age and sex are included in the model, likely due to the small sample size and strong age-NFT associations within the dataset. We have also added Supplementary Fig. 4 which shows the results of the pointwise analysis correlating thickness and summary quantitative measures of NFT burden with both age and sex included as covariates. Once again, the associations are weakened with only atrophy patterns in the CA1/subiculum region remaining statistically significant when using NFT burden measures derived from "late tau subregions".

Regarding correction for intracranial volume (ICV), unlike volumetric measurements, previous studies have suggested that cortical thickness measurements are more robust and do not need to be adjusted for brain size [1]. Furthermore, it is not common practice to correct for ICV in in vivo studies analyzing thickness of the MTL [2,3].

[1] Westman, Eric, et al. "Regional magnetic resonance imaging measures for multivariate analysis in Alzheimer's disease and mild cognitive impairment." *Brain topography* 26 (2013): 9-23.

[2] Yushkevich, Paul A., et al. "Automated volumetry and regional thickness analysis of hippocampal subfields and medial temporal cortical structures in mild cognitive impairment." *Human brain mapping* 36.1 (2015): 258-287.

[3] Xie, Long, et al. "Longitudinal atrophy in early Braak regions in preclinical Alzheimer's disease." *Human brain mapping* 41.16 (2020): 4704-4717.

Some additional minor points:

6) the discussion is quite lengthy and can be written in a more concise manner.

RE: We have tried to cut down the length of the discussion where possible while focusing on better highlighting the contributions and relevance of our findings.

7) semi-quantitative data of the contralateral side seems superfluous when you have quantitative data from the ipsilateral side, I'd suggest to move paragraph 2.3.1 to the supplement, and not have it in the results/discussion as much. Focus on the quantitative data.

RE: As suggested by the reviewer, we have tried to reduce the emphasis on the contralateral, semi-quantitative data in the discussion section and don't mention it in the abstract. However, we still believe this data would be of interest to readers within the main manuscript and have instead re-arranged the results to now present the quantitative data first, and the semi-quantitative data as a supporting analysis. The analysis using semi-quantitative data from the contra-lateral side allows us to explore structure/pathology correlations across a much larger and better-defined patient population than has been presented in prior work. Furthermore, unlike the quantitative analyses, which are limited by the smaller sample size, the larger semi-quantitative dataset allows us to correct more reliably for age and sex effects and includes analyses characterizing the effects of co-occurring TDP-43 and alpha-synuclein pathologies.

8) Figure 4 means an explanation for PP.

RE: PP refers to the perforant pathway and is defined in the caption for Figure 4. We have now also added a table (Table 2) that lists the abbreviations and definitions for the different anatomical subregions in a single place to provide more clarity.

9) the abstract is very methodological, what are the main results and conclusion?

RE: We have now revised the abstract to focus more on the results and better describe the key takeaways of this work, and shortened the length to fit the journal specifications"

Abstract: "Our current understanding of the spread and neurodegenerative effects of tau neurofibrillary tangles (NFTs) within the medial temporal lobe (MTL) during the early stages of Alzheimer's Disease (AD) is limited by the presence of confounding non-AD pathologies and the two-dimensional (2-D) nature of conventional histology studies. Here, we combine ex vivo MRI and serial histological imaging from 25 human MTL specimens, to present a detailed, 3-D characterization of quantitative NFT burden measures, in the space of a high-resolution, ex vivo atlas with cytoarchitecturally-

defined subregion labels, that can be used to inform future in vivo neuroimaging studies. Average maps show a clear anterior to poster gradient in NFT distribution, and a precise, spatial pattern with highest levels of NFTs found not just within the transentorhinal region but also the cornu ammonis (CA1) subfield. Additionally, we identify granular MTL regions where measures of neurodegeneration are likely to be linked to NFTs specifically, and thus potentially more sensitive as early AD biomarkers.”

10) I'm missing a (figure) example of how quantification is done/measured?

RE: As suggested, we have now included Supplementary Fig. 7, which provides a schematic illustration of the weakly supervised segmentation pipeline used to derive the 3D NFT burden maps. Additionally, as mentioned in a response to Reviewer 1, the methods section now includes a more detailed description of the machine learning classifier that was trained to quantify NFT burden from serial histological images, in addition to referencing Yushkevich et al., which provides more details of the quantification pipeline and additional example figures.

11) Figure 2 "frequency of NFT burden" and "average NFT burden" have overlapping color schemes in the same image?

RE: We have now changed the color map used to display the “frequency of NFT burden” so that there isn’t as much overlap in the color schemes.

12) Figure 6 "ranking in order of decreasing mean NFT burden level" is 1-6, which is similar to increasing NFT Braak staging, confusing?

RE: We recognize the lack of clarity in the current figure and have changed the labelling of the legend in Figure 6 to instead indicate decreasing levels of mean NFT burden from regions with the highest level of mean NFT burden (dark red) to subregions with the lowest level of mean NFT burden (yellow). The rank numbers have been removed to avoid confusion.

Reviewer #3 (Remarks to the Author):

Ravikumar et al. investigated the pattern of tau neurofibrillary tangles (NFTs) and its association with cortical thickness in Alzheimer's disease (AD) post-mortem brain tissues using a novel 3-D quantitative map of histological analysis of AD pathology. In this study, the authors included 55 donors aged between 57-99 years to support the overall stereotypical spreading of Braak staging although there were some diverging regions that may have been difficult to evaluate appropriately in the traditional 2-D slices of the brain tissues. Moreover, the study shows association between the NFTs and other co-pathology such as TDP-43 and alpha-synuclein to be associated with cortical thickness. The study methodology and design are elegant, and I commend all the work that went into these histological studies, ex-vivo scanning, and the mapping between the different modalities. And although the sample size may seem small for an in-vivo study, this is quite a large sample size for such ex-vivo studies. While I appreciate the advanced mapping & registration methods, the use/creation of the ex-vivo atlas, and inclusion of other co-pathologies to provide a fine-grained, comprehensive mapping of tau pathology, several issues and questions need to be addressed or further clarified.

Major comments:

1) The authors could better describe the new findings of this current paper compared to their prior recent publication (Yushkevich et al. Brain 2021). As presented, the current study seems largely an extension of their prior work with a bigger sample size, early vs. late stages along the AD continuum, and some improvements on the atlas (e.g. full extent of hippocampal SRLM layer). For example, I find the sentence in the abstract “characterize, for the first time, three-dimensional NFT spread across the human MTL at the early vs. late stages of AD” somewhat misleading as the authors already have a publication on their 3D quantification technique in the MTL. Similarly, in the discussion: “we characterize at an unprecedented level of detail, the 3-D probabilistic distribution of NFT burden at the different stages of AD”. however, what is unprecedented is (the application of their technique in) a larger cohort that encompasses the AD continuum. This paper is not describing a “novel” technique. What is new is the creation of an ex-vivo MR atlas consisting of cytoarchitecture-guided segmentations of MTL subregions, and it's not entirely clear what new insights the improved atlas and additional data are providing.

RE: We note that Reviewer 2 raised a similar concern (comment 1). In response to both these comments, we have revised the Introduction and Discussion sections of our manuscript to better describe the findings of this current paper compared to Yushkevich et al. (2021). We also focus more on describing the broader impact of our work and the potential utility of our findings towards the development of improved biomarkers and future in vivo and ex vivo studies in AD.

While we agree that the techniques used in this paper are not novel, we would like to highlight that compared to the 3D NFT mapping presented in Yushkevich et al. the current work presents a much higher resolution characterization of NFT pathology within

the MTL. In Yushkevich et al. (2021), a coarse ex vivo atlas was first constructed by applying the conventional, image-based population template building algorithm to our ex vivo scans. This rough ex vivo template was then registered to the MNI in vivo atlas for visualization of the 3D NFT burden maps. Furthermore, the anatomical regions of interest (ROIs) used for describing anatomical differences in NFT distribution were based on in vivo atlases that may not have matched individual anatomical structures well in the in vivo template space. Here, we use advanced, shape-based registration techniques to more accurately align the ex vivo scans and describe anatomical differences in NFT distribution using histology-based anatomical ROIs. We would also like to highlight that unlike our previous studies, the current study includes novel analyses linking quantitative NFT burden measures with thickness measures obtained in the same region, and we limited our analyses to cases along the AD continuum. Taken together, these various methodological improvements and the better selection of brain donors result in a far more precise and reliable characterization of NFT pathology and its neurodegenerative effects within the MTL.

Discussion pg. 14: "In Supplementary Fig. 6, we map our ex vivo atlas to the in vivo brain template used in [32] to provide a side-by-side comparison of the 3D mapping presented in the current work and [32]. Although the distribution of NFT pathology observed in the current dataset is mostly consistent with the pattern of distribution reported in [32], we see that by leveraging a more advanced, shape-based atlas construction pipeline and cytoarchitecture-guided MTL subregion segmentations derived from serial histology, the current mapping provides a fine-grained visualization of the differential involvement of NFT pathology across the MTL that is also more precisely linked to the specific subregion boundaries. While in [32], an anterior to posterior gradient in NFT distribution was observed in both the parahippocampal gyrus and hippocampus, here our average maps suggest greater NFT accumulation in the anterior parahippocampal gyrus but more extensive CA1 involvement extending to include both the anterior and posterior regions of the hippocampus, even during the early Braak stages. This difference could also be indicative of some Braak III cases being mis-classified as Braak Stage II due to NFT pathology not being present or missed in the CA 1/subiculum region of a single histology slice sampled from the opposite hemisphere during standard autopsy."

2) The current AD literature supports that the distribution of tau pathology is strongly associated with the symptoms and/or phenotypes observed in AD patients. Given that the study showed the evidence that the brain tissues investigated followed the typical spreading of tau described by Braak and Braak, is possible for the authors to provide any cognitive or behavioral metrics and associate them with the pathology?

RE: Thank you for this suggestion. We agree that it would be interesting to investigate the association between tau pathology and cognitive metrics. Currently, we don't have enough data to reliably examine this since only nine of the cases with quantitative NFT burden maps have cognitive measures available. Furthermore, our study focuses on a very specific region of the brain, largely affected during the preclinical stages of AD. As we expand our dataset and extend our quantitative characterization to other parts of the

brain, in future work we will be able to better examine patterns of tau pathology associated with cognitive or behavioral metrics.

3) Based on the description, it seems like the authors reported the main effect of NFTs, TDP-43, or a-synuclein on neurodegeneration. Were there any significant interactive effects between the different co-pathologies?

RE: In the dataset used for the semi-quantitative analyses, the average TDP-43 pathology rating is 0.16 ± 0.45 , with 81% of the cases having no TDP-43 pathology within the MTL. Similarly, the average alpha-synuclein pathology rating is 0.27 ± 0.61 , with 79% of the cases having a rating of zero. Therefore, we currently do not have a large enough dataset with sufficient pathological variability to explore interaction effects thoroughly. We now suggest investigation of interactive effects between the different co-pathologies as future work.

Discussion pg. 14: "In our analysis using semi-quantitative neuropathology ratings, when comparing atrophy patterns in relation to tau pathology versus TDP-43 pathology, we observe that TDP-43 pathology has a much greater effect on the pyramidal layers while tau is more correlated with the SRLM. This is in line with the CA1 subfield showing robust atrophy, particularly with hippocampal sclerosis associated with TDP-43 pathology [52]. These patterns of correlation suggest that the ERC and SRLM might be more specific to tau pathology. In [34], Wisse et al. found no significant associations when investigating synergistic effects between tau and TDP-43 on MTL neurodegeneration. This analysis was done in a slightly larger dataset, albeit with more variable neuropathological diagnoses. Future analyses of a larger AD dataset including more cases with severe TDP-43 pathology would enable further investigation of interactive effects between tau and TDP-43 pathology."

4) Linking NFT patterns to thickness on this high-res 3D level is fascinating but remains confined to the MTL. How do the authors see an expansion/translation of their work to larger parts of the brain? As in-vivo literature has shown neocortical tau spread along neural connections in the brain (both functionally and structurally), how could different [connected] parts of the brain be integrated within a more continuous 3D model of tau spread?

RE: We have been acquiring high-resolution, ex vivo scans of whole hemispheres and ongoing efforts are being made in the lab to develop tools and pipelines to automate cortical segmentation, accurately quantify thickness in different regions in the brain and build a template from postmortem MRI [1]. We see the present work serving as a foundational framework for quantitative 3D serial histological imaging and MRI template construction that can be expanded in future studies to functionally or structurally connected brain regions of interest. Due to the laborious nature of this undertaking, a continuous mapping across the whole hemisphere would be challenging to implement across a large number of specimens. To be able to characterize NFT/thickness associations across a wider set of anatomical regions beyond just the MTL, we are

currently in the process of obtaining neuropathology measurements from the same hemisphere histology performed in select brain regions and developing machine learning-based quantitative pathological ratings [2]. We now mention this as a future direction in the discussion section.

Discussion pg. 16: “We have also been acquiring high-resolution, ex vivo scans of whole hemispheres and ongoing efforts are being made to develop tools and pipelines to examine tau/structure associations at the level of whole hemispheres [59,60]. Therefore, the current work also serves as a foundation for future ex vivo studies, leveraging larger datasets and more expansive quantitative histological characterizations, to examine associations between pathology and cognitive metrics, connectivity patterns and other neuroimaging features, thus improving our understanding of AD biology and progression.”

[1] Khandelwal, Pulkit, et al. "Automated deep learning segmentation of high-resolution 7 T ex vivo MRI for quantitative analysis of structure-pathology correlations in neurodegenerative diseases." arXiv preprint arXiv:2303.12237 (2023).

[2] Denning, Amanda E., et al. "Evaluation of quantitative measurements of tau pathology with semi-quantitative ratings and age." *Alzheimer's & Dementia* 19 (2023): e077083.

5) Is there a mapping between the original in-vivo atlas of the prior work and the new ex-vivo atlas? It would be quite interesting to see if the NFT burden trends still hold and how they compare in the space of a (down-sampled) in-vivo atlas with (artificially) much larger voxel sizes compared to the ex-vivo atlas -mimicking an in-vivo structural MRI or PET scan.

RE: As suggested, we have now mapped the ex vivo atlas presented in this work to the original in vivo atlas used in our prior work. Supplemental Fig. 6 presents a side-by-side comparison of the average NFT burden map computed in the current work (n = 25) and the average NFT burden map computed in Yushkevich et al, both visualized in the space of the MNI in vivo atlas. To map the ex vivo template into the in vivo template, highly smoothed deformable registration was applied to MTL segmentations (consisting of hippocampus vs cortex labels) defined in both domains. While these initial results look promising to be able to provide a more accurate mapping between the ex vivo and in vivo domains, we have been acquiring in vivo and ex vivo scans from the same subject. This same-subject dataset which will enable more accurate matching of anatomical regions between the two domains and facilitate translation of the NFT burden maps into the in vivo structural MRI space.

6) Was quantitative immunohistochemistry performed in some of the tissue samples for any of the pathologies? And how did it correspond with the semi-quantitative ratings?

RE: For this study, we only performed serial histological imaging with quantitative tau immunohistochemistry using the MTL tissue that underwent MRI scanning. The semi-

quantitative ratings are performed using tissue samples obtained from the hemisphere contralateral to the one that was scanned. We have been digitally scanning the histological sections used for semi-quantitative, neuropathological evaluation. However, this data is only available for brain donors to the CNDR at UPenn. In a preliminary pilot study, we applied the machine learning model used for quantifying ipsilateral NFT burden, to the contralateral histological images, and observed high overall agreement between the derived quantitative tau measures and corresponding semi-quantitative ratings [1]. It is important to note that the histology slides used for ipsilateral analysis (AT8-stained sections) are processed differently from the histology sections used for contralateral, neuropathological evaluation (thinner, PHF1-stained sections). Furthermore, these quantitative measures are not regionally specific enough to the sampling location used for semi-quantitative analysis. Therefore, these quantitative ratings are not yet sufficiently validated for us to use in this study. Ongoing work in our lab is focused on generating pathology annotations in PHF1 slides to train and validate machine learning tools to better quantify tau pathology in our contralateral tissue samples.

[1] Denning, Amanda E., et al. "Evaluation of quantitative measurements of tau pathology with semi-quantitative ratings and age." *Alzheimer's & Dementia* 19 (2023): e077083.

Minor comments:

7) Given the possible histological examinations, have the authors investigated any neuroinflammatory or vascular biomarkers/cells?

RE: So far, we have not directly examined the presence of neuroinflammatory or vascular alterations and only focus on a subset of proteinopathies. We are aware that it is very important to investigate the neuroinflammatory and vascular components of neurodegenerative processes. The continuation of the present study will incorporate those components and investigate MRI white matter hyperintensities in association with histological samples examined with IBA-1 as microglial marker of neuroinflammation, as well as H&E and Luxol Fast Blue as general neuropathological markers. We also plan to explore other possible markers to get the best possible protocol adapted to our tissue processing procedures. We now mention this in the discussion.

Discussion pg. 16: "We also aim to develop quantitative histopathological markers of neuroinflammation and vascular disease to be able to examine their role in neurodegenerative processes. Using these rich quantitative datasets, future analyses can be performed leveraging the atlas to better characterize the relationship between tau burden, co-morbid neurodegenerative pathologies, neuronal density, and cortical thinning."

8) Is there any information on APOE status of the subjects?

RE: APOE status is only available for the subset of autopsy cases received through the Center for Neurodegenerative Disease Research at the University of Pennsylvania. This

information has been added to the extended demographics provided in Supplementary Table 1.

9) In exploratory and uncorrected analyses (for multiple corrections), it would be beneficial to perform some sort of bootstrapping or permutations and provide ranges of correlations and effect sizes or p-values.

RE: Thank you for this great suggestion. We have now included 95% confidence intervals based on bootstrapping when reporting the Spearman rank correlations between regional thickness and quantitative NFT burden measures in Figures 3 and 6.

Methods pg. 19: "Additionally, we conducted ROI analyses assessing the local relationship between NFT burden and thickness by performing one-sided, partial Spearman correlation analyses using the R package "ppcor". 95% confidence intervals were determined based on bootstrapping with 1000 iterations using the R package "boot"."

10) - Numerous neuroimaging studies suggest that the structural and/or functional connectomes may be implicated in tauopathy and its spreading. As the proposed method combine histological data with neuroimaging space, can the authors discuss how such structural and/or functional networks can be addressed?

RE: Thank you for this comment. We believe that findings and methods presented in the current study would be of immense value in improving our understanding of the role of structural and functional connectivity in the spreading of tauopathy. In the initial submission, we briefly mentioned in the discussion that the regional patterns of high NFT burden can be used to inform seeds in functional MRI analyses. We now include a more expanded discussion of how the histology-based anatomical labelling would provide more accurate ROIs for seeding connectivity analyses and facilitate studies examining intra-MTL connectivity. Furthermore, while the current study focuses on the MTL, the presented frameworks can be extended to other brain regions of interest, such as structurally or functionally connected hub regions, for more detailed histological analysis.

Discussion pg. 15: "Current studies examining the role of structural and functional connectivity in the spread of tau pathology typically use the ERC as a seed for connectivity analysis, with definitions based on in vivo segmentation protocols and tau-PET [53,54] Recent studies have explored more granular analyses associating patterns of tau deposition with connectivity networks derived from the anterior-lateral and posterior-medial ERC [53,55] In [55], Hrybouski et al. thresholded and binarized the 3D NFT burden maps presented in our earlier work [32] to define tau-based MTL ROIs as seed for in vivo analysis of MTL-AT and MTL-PM functional connectivity. Such studies of both intra-MTL and MTL-dependent connectivity would benefit from the improved 3D mapping of NFT burden, and detailed, histology-based anatomical labelling of MTL

subregions and entorhinal subfields presented in this work, to inform more accurate seed regions for connectivity analysis.”

11) SRLM should be defined in the abstract.

RE: The definition has now been added.

12) Figure 4: the pvalue of the last panel (area TH) seems incorrect.

RE: Thank you for pointing this out. This was due to a typing error. The p-value has been corrected from 0.22 to 0.022.

13) Table 1: what does the primary vs secondary diagnoses refer to?

RE: We have now included in the Results section, an explanation of how the primary versus secondary neuropathological diagnosis is determined.

Results pg. 3: “Note that in brains with co-occurring pathologies, the neuropathology that is more dominant or advanced is listed as primary neuropathological diagnosis. However, there are cases where it is not clear which neuropathology is more dominant. In such cases, the driving neuropathology that is more responsible for the clinical phenotype is listed as the primary neuropathological diagnosis even though both the “primary” and “secondary” pathological diagnoses are equally severe.”

REVIEWERS' COMMENTS

Reviewer #1 (Remarks to the Author):

The authors have addressed all my concerns in their revision.

Reviewer #2 (Remarks to the Author):

The authors have addressed all my concerns in a satisfactory manner, thank you for the extensive replies.

Reviewer #3 (Remarks to the Author):

The authors have addressed my comments. Thank you.